# Mental Modelling of Reinforcement Learning Agents by Language Models

**Wenhao Lu**     *wenhao.lu@uni-hamburg.de*
*University of Hamburg*

**Xufeng Zhao**     *xufeng.zhao@uni-hamburg.de*
*University of Hamburg*

**Josua Spisak**     *josua.spisak@uni-hamburg.de*
*University of Hamburg*

**Jae Hee Lee**     *jae.hee.lee@uni-hamburg.de*
*University of Hamburg*

**Stefan Wermter**     *stefan.wermter@uni-hamburg.de*
*University of Hamburg*

**Reviewed on OpenReview:** *https://openreview.net/forum?id=JN7iNWaPTe*

## Abstract

Can emergent language models faithfully model the intelligence of decision-making agents? Though modern language models already exhibit some reasoning ability, and theoretically can potentially express any probable distribution over tokens, it remains underexplored how the world knowledge these pre-trained models have memorized can be utilised to comprehend an agent's behaviour in the physical world. This paper empirically examines, for the first time, how well large language models (LLMs) can build a mental model of reinforcement learning (RL) agents, termed *agent mental modelling*, by reasoning about an agent's behaviour and its effect on states from agent interaction history. This research attempts to unveil the potential of leveraging LLMs for elucidating RL agent behaviour, addressing a key challenge in explainable RL. To this end, we propose specific evaluation metrics and test them on selected RL task datasets of varying complexity, reporting findings on agent mental model establishment. Our results disclose that LLMs are not yet capable of fully realising the mental modelling of agents through inference alone without further innovations. This work thus provides new insights into the capabilities and limitations of modern LLMs, highlighting that while they show promise in understanding agents with a longer history context, preexisting beliefs within LLMs about behavioural optimum and state complexity limit their ability to fully comprehend an agent's behaviour and action effects.

## 1 Introduction

Large language models (LLMs) perform surprisingly well in some types of reasoning due to their commonsense knowledge (Li et al., 2022b), including math, symbolic, and spatial reasoning (Kojima et al., 2022; Yamada et al., 2023; Momennejad et al., 2023; Zhao et al., 2024b). Still, most reasoning experiments focus on human-written text corpora (Cobbe et al., 2021; Lu et al., 2022), rather than real or simulated sequential and temporal data, such as interactions of reinforcement learning (RL) agents with physical simulators. The latter scenario unveils the potential of leveraging LLMs for elucidating RL agent behaviour, with which we may further facilitate human understanding of such behaviour—a long-standing challenge in explainable RL (Milani et al., 2024; Lu et al., 2024). Leveraging LLMs for this purpose is tempting as they can provide

explanatory reasoning over a sequence of actions in human-readable language, and this is possible due to their known ability to in-context learn from input-output pairs (Garg et al., 2022; Min et al., 2022; Li et al., 2023). This capability allows LLMs to integrate both interaction history and task-specific knowledge for more nuanced explanations, unlike most existing explainability techniques for RL (Milani et al., 2024), which have failed to consider the dynamics of agent-environment interactions.

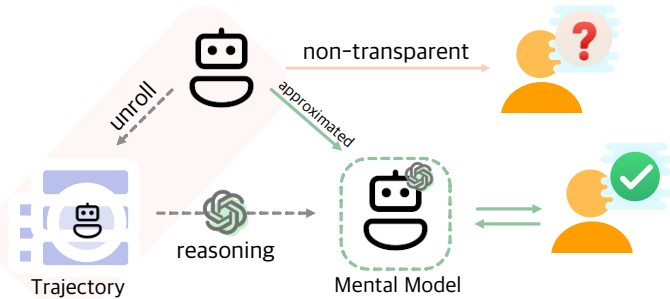

Figure 1: A conception of LLMs approximating an agent's mental model where language models reason over non-transparent agent rationale into more interpretable artefacts for end-users.

There is an ongoing debate about whether the next-token prediction paradigm of modern LLMs can model human-like intelligence (Merrill & Sabharwal, 2023; Bachmann & Nagarajan, 2024). While next-token predictors can theoretically express any conceivable token distribution, it remains underexplored how the world knowledge these models have memorised during the pre-training phase (Roberts et al., 2020) can be utilised to comprehend an agent's behaviour in the real or simulated physical world. In this work, we conduct the first empirical study to examine whether LLMs can build a mental model (Johnson-Laird, 1983; Bansal et al., 2019) of agents (Figure 1), termed *agent mental modelling*, by reasoning about an agent's behaviour and the consequences from its interaction history. Understanding LLMs' ability to interpret agent behaviour could guide the development of agent-oriented LLMs that plan and generate sequences of *embodied* actions. Though recent studies (Li et al., 2022a; Huang et al., 2023) show that LLMs can aid in planning for embodied tasks, they merely demonstrate a limited understanding of the physical world, often constrained by basic, low-level control mechanisms. Further, this agent understanding could also inform the use of LLMs as communication mediators between black-box agents and various stakeholders (see Sec. 5.3 for a discussion).

Understanding RL agent behaviour is more complex for LLMs than solving traditional reasoning tasks, which often involve the procedure of plugging different values into equations (Razeghi et al., 2022). In our study, we formalize the process of agent mental modelling, requiring LLMs to not only comprehend the actions taken by the agent but also perceive the resulting state changes (details in Sec. 3).

The contributions of this paper include: 1) we shed light on evaluating LLMs' ability to build a mental model of RL agents, including both agent behaviour and environmental dynamics, conducting quantitative and qualitative analyses of their capabilities and limitations; 2) we present empirical evaluation results in RL tasks, offering a well-designed testbed for this research with proposed evaluation metrics, and discuss the broader implications of enabling agent mental modelling.

## 2 Related Work

**Mental Modelling for Explaining Agents.** With rapid advancements in RL techniques, understanding the behaviour of black-box RL agents has become increasingly critical yet challenging (Qing et al., 2022; Milani et al., 2024). Various methods have been proposed to improve human comprehension of agent decisions, including visual explanations that highlight salient state features (Greydanus et al., 2018; Iyer et al., 2018), building surrogate models like decision trees to approximate original complex agents (Bastani et al., 2018), and learning interpretable agents through causal reasoning (Lu et al., 2024). From a psychological perspective, the explainability of RL agents relates to mental modelling, i.e., understanding how learning systems function (Johnson-Laird, 1983; Bansal et al., 2019). Previous studies often focus on isolated aspects

of agent behaviour, such as state saliency or reward importance (e.g., a salient state leading to a specific action or a reward motivating a behaviour (Juozapaitis et al., 2019)). Moreover, most techniques fail to engage with the agent's learning process or interactions with the environment. This work explores the potential of LLMs to facilitate mental modelling of agents by reasoning over agent-environment context, with the anticipation of providing more comprehensive explanations-beyond state/reward importance—such as revealing behavioural patterns and biases in human-readable text. LLMs are uniquely suited for this task due to their strong in-context learning capabilities (Brown et al., 2020), allowing them to interpret an agent's interaction history and provide richer insights into its behaviour than existing explainable RL techniques. Given their emergent capabilities (Wei et al., 2022a) (e.g., to mediate between humans and machine learning models), this paper takes a concrete step toward leveraging LLMs for more intuitive mental modelling of RL agents.

**In-Context Learning.** LLMs have exhibited strong performance in inferring answers to queries upon being given input-output pairs without gradient updates, a capability known as in-context learning (Brown et al., 2020; Garg et al., 2022; Min et al., 2022; Li et al., 2023). In this study, we focus on evaluating LLMs' understanding of agents within in-context learning but applied to sequential decision-making settings (Xu et al., 2022). Here, the context is in the form of state-action-reward tuples instead of input-output tuples. Closely related to our work is in-context reinforcement learning, where pre-trained transformer architecture-based models are fine-tuned to predict actions for query states in a task, given history interactions (Laskin et al., 2022; Lee et al., 2023; Lin et al., 2023; Wang et al., 2024). Unlike this line of work, we aim to evaluate LLMs' capability of building a mental model of RL agents via in-context learning, instead of optimising LLMs.

**Internal World Models.** LLMs can also be grounded to a specific task such as reasoning in the physical world or fine-tuned for enhanced embodied experiences (Liu et al., 2022; Xiang et al., 2023). However, because our focus is on the off-the-shelf performance of LLMs, we avoid this by creating a collection of interactions of RL agents with physics engines (e.g., MuJoCo (Todorov et al., 2012)). This results in a more challenging dataset benchmarking that does not explicitly query the LLMs for physics understanding, instead testing their inherent capability to understand the dynamics and rationale behind an RL agent's actions. This allows us to look into the inherent internal world model (Lake et al., 2017; Amos et al., 2018) of LLMs, which may offer capabilities for planning, predicting, and reasoning, as seen in works on embodied task planning (Ahn et al., 2022; Driess et al., 2023).

## 3 LLM-Xavier Evaluation Framework

Our work studies the capability of LLMs to understand and interpret RL agents, i.e., *agent mental modelling* in the context of Markov Decision Process (MDP) $\mathcal{M}$ (Puterman, 2014), including policies $\pi : \mathcal{S} \rightarrow \mathcal{A}$ and transition function $T : \mathcal{S} \times \mathcal{A} \rightarrow \mathcal{S}$, where $\mathcal{S}$ represents the state space and $\mathcal{A}$ represents the action space. See Figure 2 for an overview of the LLM-Xavier[1] evaluation framework.

### 3.1 In-Context Prompting

The evaluation is carried out in the context of an RL task $\mathcal{T}$ which can be viewed as the instantiation of an MDP $\mathcal{M}$. For each $\mathcal{T}$, we compile a dataset of interactions between the agent and the task environment, consisting of traversed state-action-reward tuples, denoted as $\mathcal{E}_{\mathcal{T}} := \{(s_i, a_i, r_i)\}_{i \leq L}$, where $L$ indicates the task episode length. Further, the subset of the interaction history with a time window (history size) $H$ ending at time $t$ is denoted as $\mathcal{E}_{t,H} := \{(s_i, a_i, r_i)\}_{t-H-1 \leq i \leq t}$, i.e., capturing the most recent $H$ tuples up to time $t \leq L - 1$.

The in-context learning prompts we constructed consist of task-specific background information, agent behaviour history, and evaluation question prompts (see Appendix B for example instantiated prompts):

  a) A system-level prompt outlining the MDP components of the environment in which the agent operates, including the state and action space, along with a brief task description.

---

[1]Inspired by Xavier from X-Men who can read minds, to signify its ability to model the mental states of RL agents.

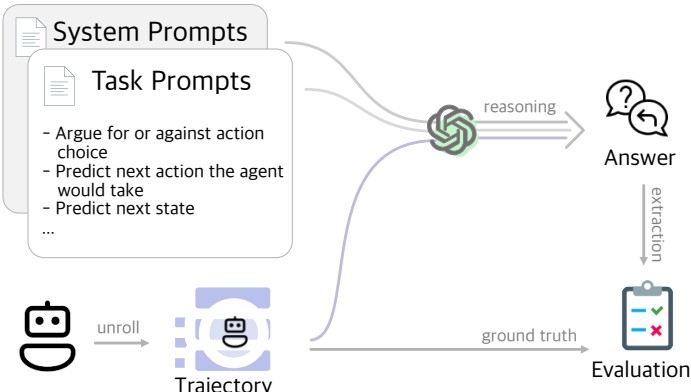

Figure 2: An overview of the LLM-Xavier workflow for offline evaluating LLMs' understanding of RL agents. The evaluation involves LLMs in-context analysing RL agent's past rollouts to argue about current actions, predict future actions, or infer state changes resulting from the agent's actions (details in Sec. 3.2).

    b) Specific prompts tailored to individual evaluation purposes (Sec. 3.2), adapted based on whether the RL setting involves a discrete or continuous state/action space.

    c) With subsets of interaction history $\mathcal{E}_{t,H}$ leading up to the current time $t$ as the in-context history, we prompt LLMs to respond to various masked-out queries $x_{\text{query}}$, corresponding to different evaluation questions, via inference over $y \leftarrow \text{LLM}(\cdot | x_{\text{query}}, \mathcal{E}_{t,H})$.

## 3.2 Evaluation Metrics

Evaluating the extent to which LLMs can develop a mental model requires examining their understanding of both the dynamics (mechanics) of environments that RL agents interact with and the rationale behind the agent's chosen actions. Mental modelling implies a comprehensive grasp of agents, which is difficult to fully capture. However, we believe that predictability is a key component of understanding. To systematically evaluate these aspects, we designed a series of targeted evaluation questions focused on predictability.

**Actions Understanding.** To assess LLMs' comprehension of the behaviour of RL agents, we evaluate their ability to accurately predict the internal strategies of agents, including

    1) *predicting next action* $y = \hat{a}_{t+1}$ given $x_{\text{query}} = s_{t+1}$,

    2) *deducing last (previous) action* $y = \hat{a}_{t+1}$ given $x_{\text{query}} = (s_{t+1}, s_{t+2})$, and

    3) *judging the next action that is given*

$$y = \begin{cases} 0 & \text{if agree with given action } a_{t+1} \\ 1 & \text{if disagree with given action } a_{t+1} \end{cases} \quad \text{given } x_{\text{query}} = s_{t+1}\,.$$

The third evaluation question assesses LLMs' reasoning by having them judge the rationale behind the agent's actions, rather than directly explaining why a specific action was taken.

**Dynamics Understanding.** To assess the awareness of LLMs to infer state transitions caused by agent actions, the evaluation of dynamics understanding includes

    (1) *predicting next state* $y = \hat{s}_{t+2}$ given $x_{\text{query}} = (s_{t+1}, a_{t+1})$, and

    (2) *deducing last (previous) state* $y = \hat{s}_{t+1}$ given $x_{\text{query}} = (a_{t+1}, s_{t+2})$.

### 3.3 Post-processing of LLMs' Predictions in Practice

The evaluation prompts are tailored to the RL tasks' action or state space. For discrete actions, LLMs predict a single integer within the action range. For continuous actions, we explore two options: (1) *predicting which bin* (from a manually divided set of 10) the next action will fall into, and (2) directly *predicting the absolute action value* within the valid range for each action dimension. For continuous state prediction, we adopt predicting relative changes (e.g., *increase*, *decrease*, *unchange*) instead of exact state values. This approach assesses the LLMs' ability to sense state transitions ($\Delta s$), e.g., changes in physical properties in physics tasks. Detailed evaluation prompts are provided in Appendix B.

We extract predictions by post-processing the generations $y \leftarrow \text{LLM}(\cdot | x_{\text{query}}, \mathcal{E}_{t,H})$ with regular expressions and compute performance by comparing them to the ground truth from the dataset. *Accuracy/matching rate* for predicting states and actions is calculated as the number of correct predictions divided by the query length ($L - \mathcal{E}_{t,H}$). Note that if LLMs predict absolute action values, both predicted and ground truth values are quantised into 10 bins, and matching is based on bin alignment. Detailed prediction post-processing is discussed in Appendix C.

## 4 Experimental Setup

We empirically evaluate contemporary open-source and proprietary LLMs on their understanding of the agent's mental model, including Llama3-8B[2], Llama3-70B, GPT-3.5[3], and GPT-4o[4] models[5]. All language models are prompted with the Chain-of-Thought (CoT) strategy (Wei et al., 2022b), explicitly encouraged to provide reasoning with explanations before jumping to the answer.

**Offline RL Datasets.** To benchmark LLMs' ability to build a mental model of an agent's behaviour, we selected a variety of tasks featuring different state spaces, action spaces, and reward spaces, resulting in a dataset comprising seven tasks (Brockman et al., 2016) with approximately 2000 query samples, represented as $(s_t, a_t, r_t)$ tuples. Four of the seven tasks are classic physical control tasks of increasing complexity, while the other three are from the Fetch environment (Plappert et al., 2018), which includes a 7-DoF arm with a two-fingered parallel gripper. Brief descriptions of each task are provided below, with corresponding visuals in Figure 3. See Appendix A.2 for detailed descriptions and Table 3 in Appendix A.1 for task statistics.

- MountainCar Task: control a car placed at the bottom of a sinusoidal valley by applying accelerations in either direction to reach the goal state on top of the right hill.

- Acrobot Task: control a two-link robotic chain to swing its free end above a specified height by applying torques at the joint.

- Pendulum Task: apply torque to swing a pendulum from a random position to an upright stance, balancing it above a fixed point.

- LunarLander Task: control a rocket's engine to land on a designated pad, using discrete actions (engine on or off), with the challenge of precise landing.

- FetchPickAndPlace Task: a 7-DoF robot moves a block to a target position on a table or in mid-air using its gripper.

- FetchPush Task: a 7-DoF robot pushes a block to a target position on a table using a locked gripper.

- FetchSlide Task: a 7-DoF robot hits a puck to slide it to a target position on a slippery table, outside its workspace.

---

[2]https://llama.meta.com/llama3/
[3]https://platform.openai.com/docs/models/gpt-3-5-turbo
[4]https://platform.openai.com/docs/models/gpt-4o
[5]Llama-3-8B-Instruct, Llama-3-70B-Instruct, gpt-3.5-turbo, and gpt-4o.

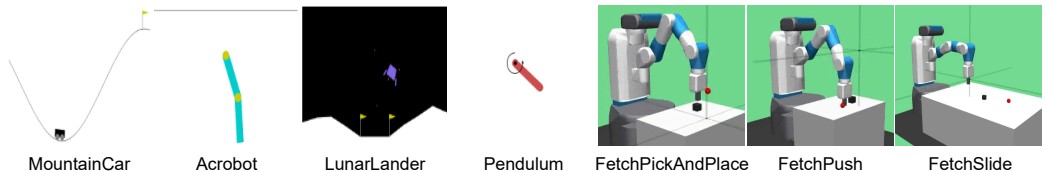

Figure 3: Visual representations of the seven tasks utilised in the evaluation experiments.

Figure 4: Comparative plots of LLMs' performance across various tasks with different history sizes (utilising **indexed history** in prompts, i.e., $\mathcal{E}_{t,H}$ presented with indices as prefixes). The Pendulum task evaluates **continuous** action prediction. First two columns show agent understanding; last two columns show dynamics understanding. A description of these tasks can be found in Appendix A.2.

# 5 Results and Discussion

## 5.1 LLMs can utilise agent history to build mental model

The experimental results indicate that LLMs can predict agent behaviours to a certain extent, for example in MountainCar, where they achieve over 75% accuracy, far surpassing the random guess baseline (1/3 chance for three action choices). However, performance declines with more challenging tasks like Acrobot and FetchPickAndPlace, as illustrated in Figure 4 (or Table 7 in Appendix F.5 for averaged accuracy) and Figure 32, which feature larger state and action spaces. We hypothesise that complex tasks require more specialized knowledge, whereas common-sense knowledge about cars and hills aids LLMs predictions in the MountainCar task.

**Longer histories enhance understanding but can degrade with excess.** We study the impact of the size of history provided in the context. As expected, as is shown in Figure 4, providing a longer history generally improves LLMs' understanding of agent behaviours. However, the benefits of including more history saturate and may even degrade, as seen in action prediction across different models, particularly Llama3-70b. This indicates that current LLMs, despite their long context length, struggle to handle excessive data in context. In this case, more data may hinder the ability to model the agent's behaviour, which is in contrast with a typical learning scenario where model performance rapidly increases as learning samples increase.

The issue of performance decline due to an excessively long history becomes more pronounced for dynamics predictions, as evidenced in the MountainCar results (refer to Figure 26 for extensive details). However, as task complexity increases, the detrimental effects of redundant history may diminish (as observed in Acrobot results in Figure 27 and Fetch-series results in Figure 32), primarily because of the challenges posed by complex state and action spaces.

On the whole, across different tasks, smaller models show significant variations in optimal history sizes for both behaviour and dynamics predictions (see Figure 4 or Table 8 in Appendix F.6 for extracted values). In contrast, larger models like GPT-3.5 are more consistent, achieving the highest accuracy with a history size of 5 (GPT-4o with a size of 1). This indicates that larger models handle contextual information more consistently in inference to an extent, while smaller models fail to utilise history statistics effectively. To further explore beyond basic accuracy, we offer a statistical analysis on how gradually increasing history size affects previous predictions (improve/worsen) in Appendix F.7.

**Regressing on absolute action values is easier than predicting action bins.** Surprisingly, LLMs perform better at predicting absolute action values than at predicting the bins into which the estimated action falls (refer to Appendix B.5 for differences in prompts). At most, LLama3-8b can allocate the numbers into categories with a mere 10.87% accuracy for the Pendulum task (GPT-3.5 achieves 39.19%), but performs better in predicting numeric values with an accuracy of up to 47.73% (GPT-3.5 scores 56.82%). A detailed comparison of the averaged accuracy across LLMs is depicted in Figure 5. We hypothesise that predicting bins requires additional math ability to categorise values using context information. Refer to Figure 39 and Appendix F.6.1 for the illustrative discrepancy.

## 5.2 LLMs' dynamics understanding has the potential to be further improved

Inferring the dynamics in a simulated world for different tasks can be challenging in many aspects, such as reasoning on a high-dimension state, computing physics consequences, and so on.

To investigate LLMs' potential of understanding dynamics, first, we investigate the impact of providing dynamics principles, which turns out to improve both behaviour and dynamics prediction when the dynamics context is informed to LLMs (see Figure 36 for details).

Further, we explicitly examined prediction performance across state components for each dimension. As depicted in Figure 6, LLMs find it relatively easier to sense car position (element 0) than velocity (element 1) for the MountainCar task; in contrast, for the Acrobot task, LLMs exhibit nearly uniform prediction accuracy across all state elements due to the difficulty in sensing state changes (see Appendix F.2 for

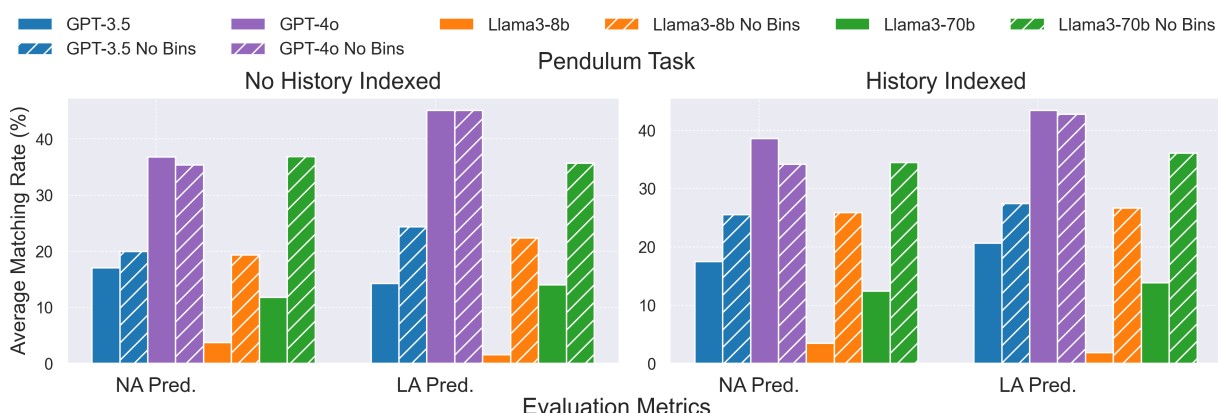

Figure 5: Comparison of models' performance in predicting *absolute action values* and *action bins* for the Pendulum task with and without indexed history in prompts. "NA. Pred." and "LA. Pred." stand for next and last action prediction, respectively, and are used throughout the paper. Hatching indicates numeric prediction accuracy ("No Bins").

details). We hypothesise that LLMs are more proficient in linear regression, as noted in Zhang et al. (2023), and the dynamics equation in MountainCar is almost linear, whereas it is non-linear in Acrobot.

Interestingly, the small model (Llama3-8b) is comparable to or even outperforms a larger model like GPT-3.5 in predicting individual state elements in some tasks, such as Acrobot. This suggests that **while small models have inferior predictive ability in actions, their understanding of action effects may not be significantly influenced by the model size, but more likely by state complexity** (e.g., predicting $y$ coordinate is easier as the lunar lander is more likely to descent in most steps). Refer to Appendix F.1 and F.2 for more illustrative results.

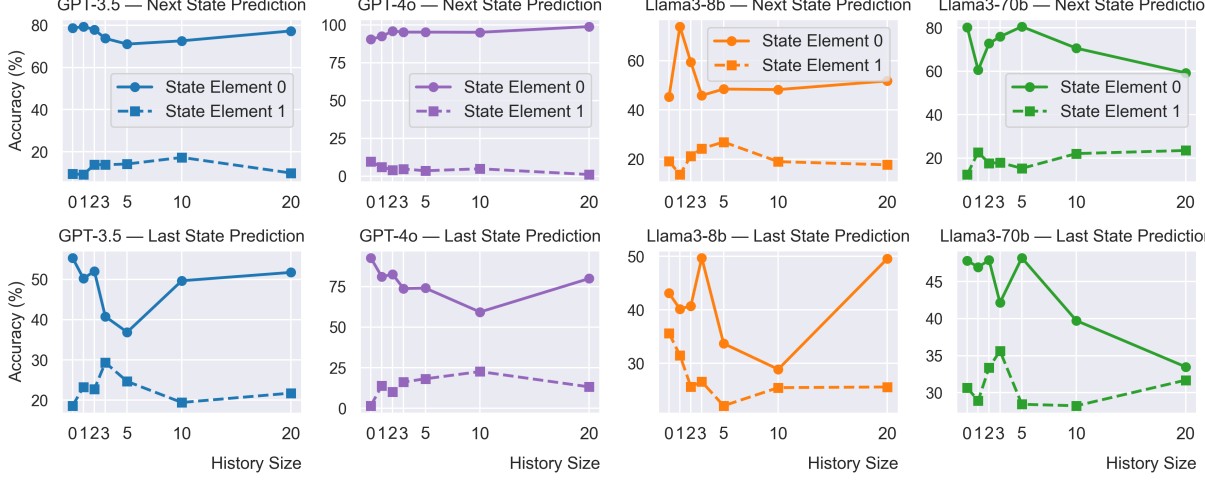

Figure 6: Dynamics of LLMs' performance on predicting individual state element for the MountainCar task (with **indexed history** in prompts).

**State span is statistically influential in LLMs' dynamics understanding.** Based on the observations above, we examined the factors that statistically influence LLMs' perception of state changes. Figure 7 demonstrates a decent linear relationship between the span of state elements and LLMs' ability to detect these changes in tasks like MountainCar, Acrobot, and LunarLander, especially with GPT-3.5. For Acrobot, we observe a low slope with a moderate fit, while LunarLander shows a positive slope with a moderate

fit. Our hypothesis posits that in a given task, a smaller state span makes sensing changes difficult due to the minimal differences in state values. Conversely, a larger span tends to improve prediction accuracy as it amplifies the discrepancies between state values, making changes more discernible. Further details and discussions are provided in Appendix F.3.

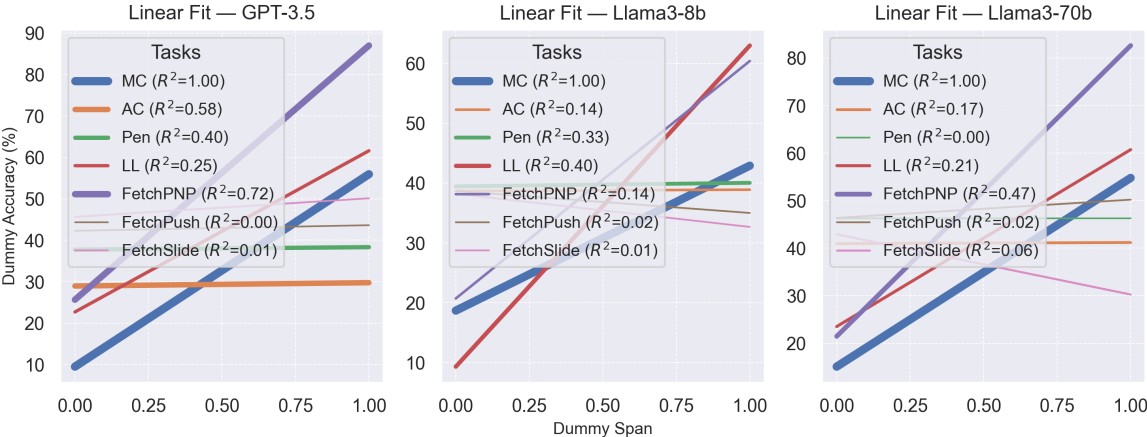

Figure 7: The $R^2$ values (also reflected by the linewidth) and slopes of the fitted linear lines for all models across various tasks. A standardised dummy state span (0-1) and accuracy scale (0-100) are utilised to account for differing state spans among tasks in the analysis. "MC" stands for MountainCar, "AC" for Acrobot, "Pen" for Pedulum, "LL" for LunarLander, "FetchPNP" for FetchPickAndPlace.

### 5.3 Understanding error occurs from various aspects

With the anticipation that LLMs' explanatory reasoning (elicited via CoT) can benefit the human understanding of agent behaviour, in addition to the existing quantitative results, we further examined the reasoning error types across LLMs *by manually reviewing their judgments on the rationale of actions taken.* Table 1 shows an examination of the MountainCar task, highlighting that LLama3-8b displays the most errors. Meanwhile, GPT-3.5, despite having superior task comprehension (e.g., referring to momentum strategies), is less effective at retaining task descriptions in memory compared to Llama3-70b. Notably, GPT-4o's responses are of higher quality overall. While GPT-4o is occasionally inconsistent with the optimal momentum strategy (i.e., initially accelerating left), it displayed a significantly better grasp of this strategy than other models. Furthermore, it avoids fundamental mistakes such as misinterpreting numbers or the presented information. Detailed error type reports are available in Appendix G.1.

Table 1: Error counts in LLMs' responses for MountainCar task over 50 steps, with various error types. Note: Multiple error types can occur within a single step.

| Error Types | GPT-3.5 | Llama3-8b | Llama3-70b | GPT-4o |
|---|---|---|---|---|
| (1) Task Understanding | 9 | 30 | 16 | **3** |
| (2) Logic | 5 | 19 | 4 | **0** |
| (3) History Understanding | 3 | 18 | 4 | **0** |
| (4) Physical Understanding | 1 | 2 | 3 | **0** |
| (5) Mathematical Understanding | 2 | 25 | 2 | **0** |
| (6) Missing Information | 9 | 10 | 13 | **6** |

**Confirmation bias in LLMs: a likely error from initial assumptions of RL agent's optimality.**
In the manual review, we queried LLMs to judge a possible next action (see corresponding prompt in

Appendix B.4) given the history of the last three actions and states, a history size determined to be optimal for most models (see Table 8). The provided next action was sometimes correct (if it was the agent's action) and sometimes incorrect, ensuring LLMs made context-based conclusions rather than merely agreeing or disagreeing with the prompt. We evaluated whether the LLMs' judgments were correct according to a human reviewer, independent of the RL agent's action correctness. An automatic evaluation compared LLMs' decisions to the RL agent's actions (as described in Sec. 3.2).

The manual evaluation did differ from the automatic evaluation, as shown in Table 2. The table's percentages refer to the proportion of LLMs responses deemed correct. This difference stems from considering a different action ground truth since the RL agent occasionally acts illogically, leading to the human reviewer deeming those actions incorrect, while automatic evaluation considers them correct from the RL agent's perspective.

Hypothetically, this difference may arise from LLMs forming mental models of actions they believe an optimal RL agent should take. This reveals another type of understanding error, where LLMs' responses may be influenced by preexisting beliefs or assumptions about optimal behaviour. As a result, LLMs sometimes propose more optimal actions than the RL agent, particularly in larger models (evidenced by the increased accuracy in human evaluation), where these beliefs in optimality might be more strongly reinforced. Exploring whether such beliefs exist in LLMs could be a promising research direction, potentially benefiting from advanced explainability techniques for LLMs Zhao et al. (2024a).

Despite these differences, the variation is minimal and not statistically significant (paired t-test, $p$-value $= 0.45 > 0.05$), suggesting that the RL agent's behaviour is near-optimal or acceptable to the human evaluator. Thus, LLMs' action judgments may offer (less biased) explanations for RL agent actions; when agent actions are sub-optimal, LLMs may help explain why those actions are deemed inappropriate.

Table 2: The accuracy of models evaluated manually or automatically for 50 steps in the MountainCar task with the metric *judging next action*.

| Model | Manual | Automatic |
|---|---|---|
| GPT-3.5 | 60% | 67% |
| Llama3-8b | 40% | 52% |
| Llama3-70b | 67% | 65% |
| GPT-4o | **85%** | **81%** |

**Pre-training biases may affect understanding despite history length.** Interestingly, we found that LLMs prediction errors may be influenced by pre-training biases towards cautious actions. Figure 33 in Appendix F.7 shows that increased history size does not affect late-episode steps where RL agents consistently accelerate right to finish. Instead, LLMs mentally "suggest" conservative actions, like accelerating left to avoid overshooting, regardless of history context size. We hypothesise that these LLMs' prediction errors may not be solely due to difficulty in processing extensive histories but also stem from inherent biases about agent behaviour and environment context, consistent with our earlier observations on confirmation bias.

## 5.4 Data format influences understanding

Prompting format generally has an impact on LLMs' reasoning performance. In the context of agent understanding, we do an ablation study to investigate the robustness of prompts on the *history format* and provided *information*. We find that:

1) Excluding the sequential indices from the history context in prompts for LLMs generally negatively impacts their performance in most tasks, indicating that LLMs still struggle to process raw data and indexing helps. The resulting performance variations are reported in Figure 8.

2) Task description, despite not being directly relevant to numerical value regression as in statistics, is essential for a better understanding of both agent behaviour and dynamics, which brings the promise of utilising LLMs to digest additional information beyond mere numerical regression when mental modelling agents. The ablation results can be found in Appendix F.8.2.

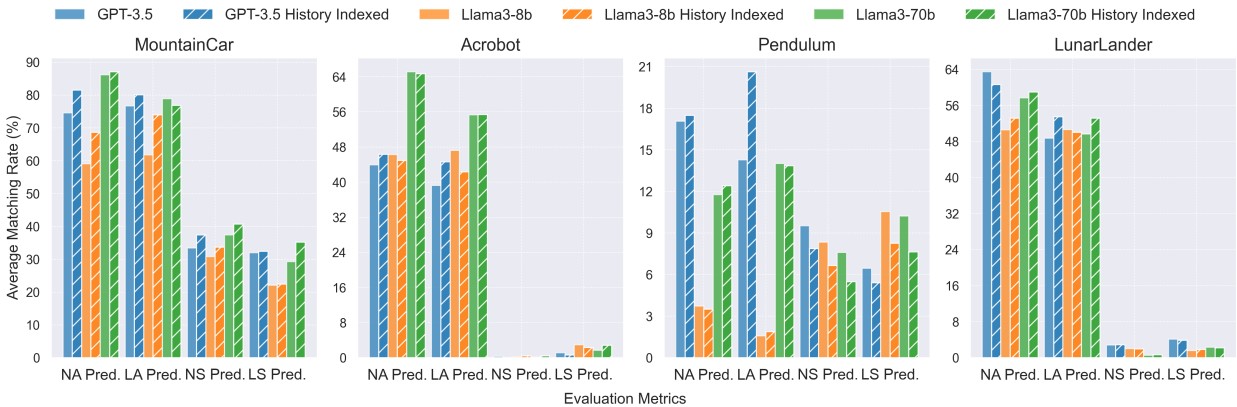

Figure 8: Performance comparison of language models with and without indexed history in prompts on various tasks. Bars with hatching indicate accuracy with **indexed history** in prompts. NS Pred. = Next State Prediction, LS Pred. = Last State Prediction.

# 6 Conclusion

This work studies an underexplored aspect of next-token predictors, with a focus on whether LLMs can build a mental model of agents. We proposed specific prompts to evaluate this capability. Quantitative evaluation results disclose that LLMs can establish agent mental models to some extent only since their understanding of state changes may diminish with increasing task complexity (e.g., high-dim spaces); their interpretation of agent behaviours may tumble for tasks with continuous actions. Analysis of evaluation prompts reveals that their content and structure, such as history size, task instructions, and data format are crucial for the effective establishment, indicating areas for future improvement. A further review of LLMs error responses (elicited via CoT prompting) highlights qualitative differences in LLMs' understanding performance, with models like GPT-4o showing superior comprehension and fewer errors compared to the small Llama3 model. Additionally, it reveals that LLM responses may be influenced by preexisting beliefs about optimal behaviour, which can be a hurdle or a catalyzer for understanding. These findings suggest the potential and limitations of in-context mental modelling of agents within MDP frameworks and underscore the possible role of LLMs as communication mediators between black-box agents and stakeholders, pointing to future research avenues.

## Potential Direction

It remains unclear whether LLMs can benefit from thousands of agent trajectories compared to the limited number of examples studied in this paper. We hypothesise that large amounts of demonstrations (state-action-reward tuples) in the prompt could enhance the capacity that LLMs have already developed. Additionally, fine-tuning LLMs with demonstrations (Lin et al., 2023; Wang et al., 2024) from specific domains may further improve their understanding capacity in these domains. Further analysis on this aspect is left for future work.

At this stage, LLMs lack several key aspects required for true understanding, often due to reasoning errors and limitations in predictability. Relying on LLMs for mental modelling without oversight could be problematic. Therefore, our key takeaway is that rather than using a fully automated, LLM-driven evaluation system for agent mental modelling, domain experts are expected to be involved upfront to audit LLMs' reasoning and ensure reasonable predictability and reliability before presenting the results to non-expert practitioners. A few concrete suggestions for utilising LLMs in agent mental modelling are provided in Appendix H.

We recognise that the issue of hallucination may exist. To increase the robustness and reliability of using LLMs for explaining an agent's behaviour, a detailed analysis of this behaviour is necessary before being deployed to a setting where they directly interact with humans. Also, our evaluation results underscore the need for developing methods to mitigate hallucinations.

Our evaluation framework can be adapted for non-deterministic MDPs with minor adjustments. In discrete state spaces (e.g., FrozenLake (Brockman et al., 2016)), LLMs can output a distribution over possible next state values, with multiple queries needed to approximate the distribution, evaluated using metrics like KL-divergence. Alternatively, LLMs could rank next state values, compared to the ground truth probabilities derived from simulator. In continuous state spaces, states can be quantised into bins, and LLMs can output a distribution or ranking over sampled values from each bin. For multi-dimensional states, distributions or rankings can be generated for each dimension separately. This approach also applies to predicting actions, where the ground truth differs by algorithm: for PPO (Schulman et al., 2017), it would be the mean and variance of the Gaussian policy, and for DQN (Hasselt et al., 2016), the Q-values for discrete actions. In continuous action spaces, LLMs can output a mean and deviation for a Gaussian distribution, which can be compared to the learned (Gaussian) distribution of the PPO agent.

Our study provides a macro-level analysis by examining the average model performance over multiple RL datasets of varying types. However, the capability of LLMs to build a mental model of agents may vary across different datasets. While our analysis discusses this aspect, it is important to explore ways of standardising this type of benchmarking for language models, which may evolve as LLMs become more intelligent. A long-term goal of this research is to facilitate human understanding of more intelligent agents in critical domains, and we see this work as a foundational step towards developing progressively more agent-oriented language models with realistic world models in mind.

Our experiments are limited to uni-modal RL tasks (i.e., using proprioceptive states), but extending them to multi-modal tasks (e.g., incorporating vision, auditory, and touch feedback) is straightforward. Multi-modal inputs can provide LLMs with richer environmental information than state vectors, and we hypothesise that these additional signals may enhance LLMs' agent mental modelling.

### Future-Proofing the Approach for LLM-Based Agents

While our current framework focuses on evaluating LLM's capability of mental modelling traditional RL agents, it remains equally relevant as RL agents evolve into LLM-based counterparts that can provide human-readable justifications for their decisions. One might argue that these self-generated explanations reduce the need for external evaluation. However, these internal justifications may not reflect a truly objective mental model as they can be shaped by the agent's underlying optimisation processes, biases, and emergent behaviours. To address this, when evaluating LLM-based agents, we can extend our framework to compare their self-reported rationales against the mental model formed by an independent evaluator LLM.

Furthermore, as LLM-driven agents rapidly adapt policies in response to dialogue histories, external instructions, or internal reasoning steps, we can integrate temporal tracking of these shifts in real-time. This enables the evaluator LLM to detect subtle policy shifts and predict future actions based on evolving rationales. As LLM-based agents expand into multi-domain environments with non-traditional state, action, and reward structures—such as language-based tasks—our framework can similarly evolve, challenging the evaluator LLM to generalise beyond conventional RL dynamics. Finally, as we leveraged human input for error-type annotation and analysis, future-proofing involves integrating semi-automated tools to handle intricate LLM-agent interactions at scale. Human evaluators may focus on higher-level interpretation and strategic insights, while automated metrics track the consistency and coherence of evolving mental models over time.

### Broader Impact Statement

We do not anticipate any immediate ethical or societal implications from our research. However, since we explore LLMs applications for enhancing human understanding of agents, it is important to be cautious about the potential for fabricated or inaccurate claims in LLMs' explanatory responses, which may arise from misinformation and hallucinations inherent to the LLMs employed. It is recommended to use our proposed evaluation prompts and task dataset with care and mindfulness.

## Acknowledgements

This research was funded by the Federal Ministry for Economic Affairs and Climate Action (BMWK) under the Federal Aviation Research Programme (LuFO), Projekt VeriKAS (20X1905)

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

# A    Statistics of Our Offline-RL Datasets

## A.1    Data Collection

The dataset of interaction histories (episodes) is collected by running RL agents in each task. Unlike Liu et al. (2022), whose physics alignment dataset contains text-based physical reasoning questions resembling physics textbooks, our dataset comprises interactions of RL agents with various physics engines (environments). For each task, episodic histories are collected by running single-task RL algorithms (Lillicrap et al., 2015; Hasselt et al., 2016; Schulman et al., 2017) to solve that task. An overview of the task dataset statistics is provided in Table 3.

Table 3: A statistical overview of the task dataset tested in the experiment.

| Tasks | # of episodes | Length per episode | State Space | State Dim | Action Space | Action Dim |
|---|---|---|---|---|---|---|
| MountainCar | 5 | $\sim 100$ | continuous | 2 | discrete | 1 (3 choices) |
| Acrobot | 3 | $\sim 100$ | continuous | 6 | discrete | 1 (3 choices) |
| LunarLander | 3 | $\sim 250$ | continuous | 8 | discrete | 1 (4 choices) |
| Pendulum | 3 | $\sim 50$ | continuous | 3 | continuous | 1 |
| InvertedDoublePendulum | 3 | $\sim 50$ | continuous | 11 | continuous | 1 |
| FetchPickAndPlace | 10 | $\sim 10$ | continuous | 25 | continuous | 4 |
| FetchPush | 10 | $\sim 10$ | continuous | 25 | continuous | 4 |
| FetchSlide | 10 | $\sim 25$ | continuous | 25 | continuous | 4 |

## A.2    A Full Task Description

Below, we provide a complete description of the MountainCar task (at the last of the section), including its MDP components. For the remaining tasks, only the task descriptions are provided. Most of the texts are credited to https://gymnasium.farama.org/.

**Acrobot Task Description.** — *The Acrobot environment is based on Sutton's work in "Generalization in Reinforcement Learning: Successful Examples Using Sparse Coarse Coding" and Sutton and Barto's book. The system consists of two links connected linearly to form a chain, with one end of the chain fixed. The joint between the two links is actuated. The goal is to apply torques on the actuated joint to swing the free end of the outer-link above a given height while starting from the initial state of hanging downwards.*

**Pendulum Task Description.** — *The inverted pendulum swingup problem is based on the classic problem in control theory. The system consists of a pendulum attached at one end to a fixed point, and the other end being free. The pendulum starts in a random position and the goal is to apply torque on the free end to swing it into an upright position, with its center of gravity right above the fixed point.*

**LunarLander Task Description.** — *This environment is a classic rocket trajectory optimization problem. According to Pontryagin's maximum principle, it is optimal to fire the engine at full throttle or turn it off. This is the reason why this environment has discrete actions: engine on or off. The landing pad is always at coordinates (0,0). The coordinates are the first two numbers in the state vector. Landing outside of the landing pad is possible. Fuel is infinite, so an agent can learn to fly and then land on its first attempt.*

**FetchPickAndPlace Task Description.** — *The task in the environment is for a manipulator to move a block to a target position on top of a table or in mid-air. The robot is a 7-DoF Fetch Mobile Manipulator with a two-fingered parallel gripper (i.e., end effector). The robot is controlled by small displacements of the gripper in Cartesian coordinates and the inverse kinematics are computed internally by the MuJoCo framework. The gripper can be opened or closed in order to perform the grasping operation of pick and place. The task is also continuing which means that the robot has to maintain the block in the target position for an indefinite period of time.*

**FetchSlide Task Description.** — *The task in the environment is for a manipulator to hit a puck in order to reach a target position on top of a long and slippery table. The table has a low friction coefficient in*

*order to make it slippery for the puck to slide and be able to reach the target position which is outside of the robot's workspace. The robot is a 7-DoF Fetch Mobile Manipulator with a two-fingered parallel gripper (i.e., end effector). The robot is controlled by small displacements of the gripper in Cartesian coordinates and the inverse kinematics are computed internally by the MuJoCo framework. The gripper is locked in a closed configuration since the puck doesn't need to be graspped. The task is also continuing which means that the robot has to maintain the puck in the target position for an indefinite period of time.*

**FetchPush Task Description.** — *The task in the environment is for a manipulator to move a block to a target position on top of a table by pushing with its gripper. The robot is a 7-DoF Fetch Mobile Manipulator with a two-fingered parallel gripper (i.e., end effector). The robot is controlled by small displacements of the gripper in Cartesian coordinates and the inverse kinematics are computed internally by the MuJoCo framework. The gripper is locked in a closed configuration in order to perform the push task. The task is also continuing which means that the robot has to maintain the block in the target position for an indefinite period of time.*

---

**MountainCar Task Prompt**

```
task_description =
The Mountain Car MDP is a deterministic MDP that consists of a car placed stochastically at
↪   the bottom of a sinusoidal valley, with the only possible actions being the
↪   accelerations that can be applied to the car in either direction. The goal of the MDP is
↪   to strategically accelerate the car to reach the goal state on top of the right hill.

observation_space =
The observation is a ndarray with shape (2,) where the elements correspond to the following:
position of the car along the x-axis (range from -1.2 to 0.6), velocity of the car (range
↪   from -0.07 to 0.07)

action_space =
There are 3 discrete deterministic actions,
0: Accelerate to the left
1: Do not accelerate
2: Accelerate to the right

reward_space =
The goal is to reach the flag placed on top of the right hill as quickly as possible, as
↪   such the agent is penalised with a reward of -1 for each timestep.

transition_dynamics =
Given an action, the mountain car follows the following transition dynamics,
velocity_t+1 = velocity_t + (action - 1) * force - cos(3 * position_t) * gravity
position_t+1 = position_t + velocity_t+1
where force = 0.001 and gravity = 0.0025. The collisions at either end are inelastic with
↪   the velocity set to 0 upon collision with the wall.

init_state =
The position of the car is assigned a uniform random value in [-0.6, -0.4]. The starting
↪   velocity of the car is always assigned to 0.

termination =
The episode ends if the position of the car is greater than or equal to 0.5 (the goal
↪   position on top of the right hill).
```

---

## B   Prompt Examples

The structured input template used for querying LLMs consists of a system prompt containing the task description, MDP components, and a prompt with specific evaluation questions, as shown in B.1 and B.2,

respectively. An example prompt for *predicting the next action* for tasks with discrete action space is depicted in B.3, and for judging agent actions in B.4. The prompts for each evaluation metric may vary slightly depending on the task type (i.e., state and action space as illustrated in Table 4), detailed in B.5.

### B.1 System Prompt

---

**System Prompt**

Below is a description of the {task_name} task.
Task description:

{task_description}

Observation space:

{observation_space}

Action space:

{action_space}

Reward space:

{reward_space}

Transition dynamics:

{transition_dynamics}

Initial state:

{init_state}

Termination:

{termination}

---

### B.2 Offline Evaluation Prompt

---

**Offline Evaluation Prompt**

Given a snippet of an episode (generated by a reinforcement learning agent optimally trained for solving the given task) of
the states:

{states}

the corresponding actions taken by the RL agent,

{actions}

and the rewards received:

{rewards}

Your task is to analyse the sequence of states, actions, and rewards to address the question:

{question}

---

### B.3 Example Next Action Prediction Prompt

```
Next Discrete Action Prediction Prompt

In next step {i} (indexed from 0), the agent transitioned to the state s{i} =
↪  {state}. Based on your observation and understanding of the agent's behaviour,
↪  can you predict the action a{i} (an integer from the given range) the RL agent
↪  will most likely take at step {i}?

Please first provide a compact reasoning before your answer to the action choice.
↪  Think step by step and use the following template in your provided answer:

1. [Reasoning]:
2. [Prediction]:
3. [Formatting]:
    Return a list with the following example format,
    ```python
    # final action choice is 0
    action_choice = [0]
    ```
Please choose only one action, even if multiple actions seem possible.
```

### B.4 Example Action Judgment Prompt

```
Action Judgment Prompt

In next step {i} (indexed from 0), the agent transitioned to the state s{i} =
↪  {state}. In this state s{i}, the agent chose to take action a{i} = {action}.
↪  Based on your observation and understanding of the agent's behaviour up to this
↪  point, critically evaluate the rationale behind the agent's choice of action
↪  a{i} in state s{i}. You're encouraged to scrutinize the correctness of the
↪  action a{i}, especially if your analysis suggests that the action might be
↪  flawed or suboptimal.

After your evaluation, determine whether to accept or reject the agent's action
↪  a{i}. Think step by step and use the following template in your provided
↪  answer:

1. [Reasoning]:
2. [Justification]: Critique whether the action is correct given the historical
↪  context and the state s{i}, and whether there are any reasons to doubt its
↪  correctness.
3. [Formatting]:
    Final vote is [True] (i.e., argue for the agent's action) or [False] (i.e.,
    ↪  argue against the agent's action)
    Return a list with the following example format,
    ```python
    # final vote is to argue against the agent's action
    final_vote = [False]
    ```
Please choose either True or False, even in cases of uncertainty or multiple
↪  possibilities.
```

Table 4: Different state spaces and action spaces of MDPs (tasks) considered in the experiments.

|  | continuous action | discrete action |
|---|:---:|:---:|
| continuous state | ✓ | ✓ |
| discrete state | ✗ | ✓ |

### B.5 Evaluation Prompts in Practice

The evaluation prompts (parts b, c in Sec. 3.1) are adapted based on the nature of the RL tasks, specifically the type of action or state space (discrete or continuous). For tasks with discrete action spaces, LLMs are prompted to output a single integer within the action range. For tasks with **continuous actions**, we evaluate two options:

- *Predicting bins*: The action range is manually divided into 10 bins, and LLMs are queried to predict which bin the RL agent's next action will fall into.

- *Predicting absolute numbers*: LLMs are queried to directly output the exact action value within the valid action range for each dimension of the action space.

For tasks involving **continuous state** prediction, we adopt predicting relative changes (e.g., *increase*, *decrease*, *unchange*) instead of exact state values. This approach assesses the LLMs' ability to sense state transitions ($\Delta s$), e.g., changes in physical properties in physics tasks.

## C   Post-processing LLMs' Predictions

We evaluate LLMs using metrics that require predicting states and actions. We extract LLMs' responses through pattern matching and compute evaluation results by comparing them with the ground truth state-action pairs from the episodes on which the LLMs are evaluated.

For predicting **discrete actions**, we compute the matching rate of the LLMs' predicted actions with the ground truth. For predicting **continuous actions**, if LLMs are prompted to *predict bins*, we compute the matching rate as we did for discrete actions, with the ground truth represented by the bin index to which it belongs. However, if LLMs are queried to directly *predict absolute action values*, we quantize both the predicted and ground truth values into bins (by dividing the original action range into 10 bins) and then measure whether they fall into the same bin.

For predicting **continuous states**, we evaluate if LLMs correctly predict the change in state, $\Delta s$, categorising increases as 1, decreases as 0, and unchanged as 2. We then compute the accuracy classification score for their predictions. We also record the accuracy of predicting changes in individual state elements, $\Delta s_i$.

## D   Pseudo-code for Performing Evaluation Metrics

### D.1   Pseudo-code for Predicting Next Action

Algorithm 1 presents an example pseudo-code for the next action prediction tasks.

## E   Types of LLMs' Understanding Failures

Table 5 lists error types we located by analysing LLMs' responses to evaluation questions for the MountainCar task. The MountainCar task is an RL task in which the agent has to correctly steer a car to move it onto a mountain with the actions: "Left Acceleration", "Right Acceleration", and "No Acceleration".

---

**Algorithm 1:** Offline Evaluation of LLMs' Agent Understanding

---

**Data:** Load offline RL dataset $\mathcal{E}_{\mathcal{T}}$ for a task $\mathcal{T}$
**Input:** Load the LLM model $LLM$
**Result:** Evaluation results
Initialize action matching counter $N \leftarrow 0$ ;
Set the history size $H$ ;
Set the maximum time steps $T_{max}$ ;
**for** $t \leftarrow 1$ **to** $T_{max}$ **do**
    **if** $t > H$ **then**
        Extract the last $H$ transitions $\mathcal{E}_{t,H} \leftarrow (s_i, a_i, s_{i+1}, r_i)_{t-H+1}^{t}$ ;
        `// Prepare input for` $LLM$ `including current state and history` $\mathcal{E}_{t,H}$
        Predict $\hat{a}_{t+1} \leftarrow LLM(s_{t+1}, \mathcal{E}_{t,H})$ ;
        **if** $\hat{a}_{t+1} = a_{t+1}$ **then**
            Increment counter $N \leftarrow N + 1$ ;
    **if** *Task goal is achieved* **then**
        **break** ;

Compute evaluation results using the counter $N$ and the query length, e.g., $T_{max} - H$ ;

---

Table 5: List of error types in LLMs' responses

| No. | Error Types |
|-----|-------------|
| (1) | Task Understanding |
| (2) | Logic |
| (3) | History Understanding |
| (4) | Physical Understanding |
| (5) | Mathematical Understanding |
| (6) | Missing Information |

Examples of each error type from various LLMs responses:

- **Task Understanding**

  **Llama3-8b** replies —

  *"However, one could argue that the agent's velocity is already quite high ([0.0075]), and accelerating further might not be necessary or even counterproductive if it leads to overshooting the goal position. Additionally, the agent's position is still relatively far from the goal ([0.5729] vs. 0.5), so it may need to adjust its strategy soon."*

  **Analysis**: In this example, the model makes multiple mistakes, it misunderstands the task as it thinks overshooting the goal would be bad, and it also wrongly interprets the state -0.5729 as 0.5729.

- **Logic**

  **Llama3-70b** replies —

  *"Given the current state s20, it's unclear why the agent would choose to accelerate to the right, as the position is still below the goal and the velocity is not high enough to guarantee reaching the goal."*

  **Analysis**: Here the reasoning logic does not make sense, If the goal is getting to the right, accelerating to the right should make sense especially if the velocity is not high enough yet-

  **Llama3-8b** replies —

  *"This pattern suggests that the agent might be trying to control the car's movement by alternating between accelerating left (action 0) and not accelerating (action 1) to navigate the valley towards the goal position on the right hill."*

**Analysis**: LLM argues that accelerating left will navigate the car toward the right.

- **History Understanding**

  **GPT-3.5** replies, given a history of actions: $a_{20} = 2, a_{21} = 0, a_{22} = 0, a_{23} = 0$ —

  *"The agent has been moving towards the right hill, and in the previous steps, it has been accelerating to the right (action 2) to gain momentum."*

  **Analysis**: The LLM claims the agent has been accelerating to the right using action 2 when the last three actions were action 0.

- **Physical Understanding**

  **Llama3-70b** replies —

  *"Given the agent's behaviour of accelerating to the right when the car is far from the right hill and not accelerating when it is closer, the choice of action 1 (no acceleration) in state s18 seems reasonable. This action allows the car to conserve momentum and potentially reach the goal position more efficiently."*

  **Analysis**: Here the LLM wrongly believes that not accelerating could allow the car to conserve momentum, however moving upwards gravity will cause the car to lose momentum should it stop accelerating.

  **GPT-3.5** replies —

  *"The agent seems to be attempting to move towards the right hill by repeatedly accelerating to the left."*

  **Analysis**: LLM doesn't realize that accelerating to the left won't move the car to the right.

- **Mathematical Understanding**

  **Llama3-8b** replies, given the state history $s_{25} = \begin{bmatrix} -4.0838 \times 10^{-1} & 1.7447 \times 10^{-3} \end{bmatrix}$, $s_{26} = \begin{bmatrix} -4.0849 \times 10^{-1} & -1.0230 \times 10^{-4} \end{bmatrix}$, and the new state $s_{27} = \begin{bmatrix} -4.104 \times 10^{-1} & -1.9 \times 10^{-3} \end{bmatrix}$ —

  *"Looking at the sequence of states provided, the car is moving to the right (position is increasing) while the velocity is decreasing."*

  **Analysis**: The LLM does not realize that the position is decreasing, moving to the left as it wrongly interprets the numbers.

- **Missing Information**

  **Llama3-8b** replies —

  *"The action of not accelerating might delay the agent's arrival at the goal position, especially when it is very close to the goal. It is crucial for the agent to maintain its momentum and continue accelerating towards the goal to minimize the time taken to reach the flag."*

  **Analysis**: The car needs to accelerate to the left to get to a position from which it can build enough momentum towards the right to overcome the right hill. The LLM is missing the information about the environment that would allow it to understand this behaviour.

  **GPT-4o** replies —

  *"Given the car's current state, taking action '0' (accelerate to the left) might not be the most optimal choice. The car is already moving left with a very small velocity, and further accelerating to the left might not significantly help in building the necessary momentum to move rightward. Instead, it might be more beneficial to start accelerating to the right (action '2') to begin the process of climbing the right hill."*

  **Analysis**: The LLM is missing the exact information about how much velocity is needed at which position to accurately judge when the agent should switch back to accelerating rightward.

# F    Additional Results: Evaluating LLMs' Understanding Across Various Tasks

## F.1    State Element Prediction Accuracy with Increased History

In the task of predicting (full) states, we also plot the prediction accuracy for individual state elements and how they vary with increased history size for different tasks: Figure 9 for the Pendulum task, Figure 10 for the Acrobot task, and Figure 11 for the LunarLander task.

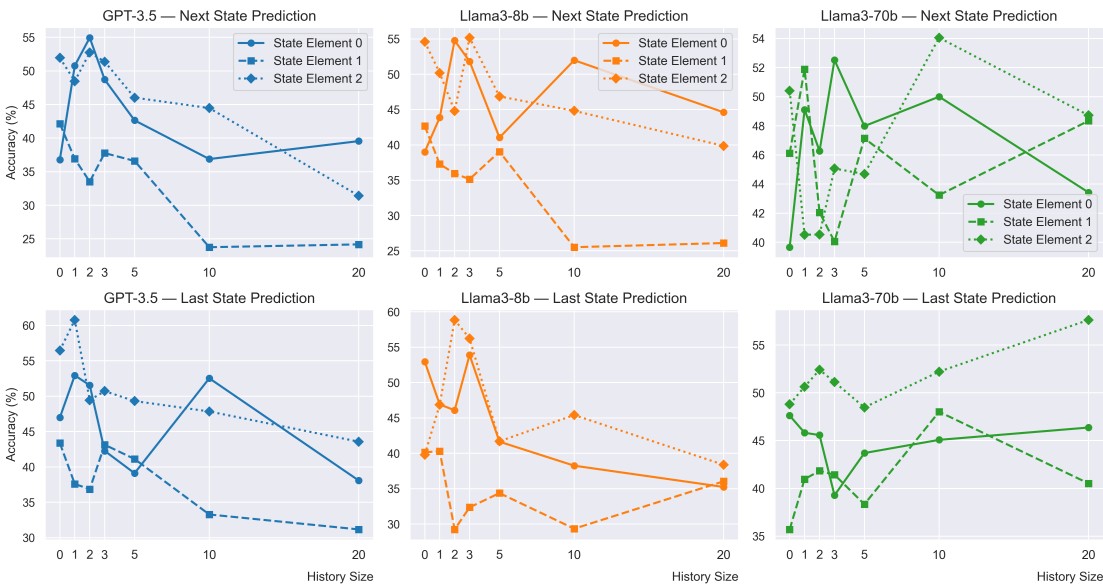

Figure 9: Dynamics of LLMs' performance on predicting individual state element for the **Pendulum** task (with indexed history in prompts).

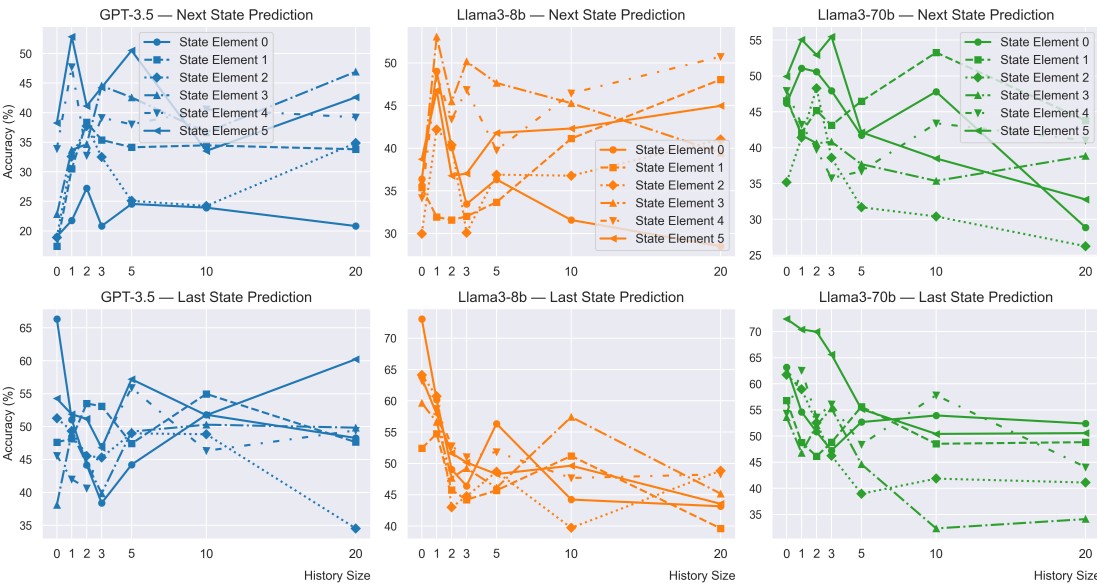

Figure 10: Dynamics of LLMs' performance on predicting individual state element for the **Acrobot** task (with indexed history in prompts).

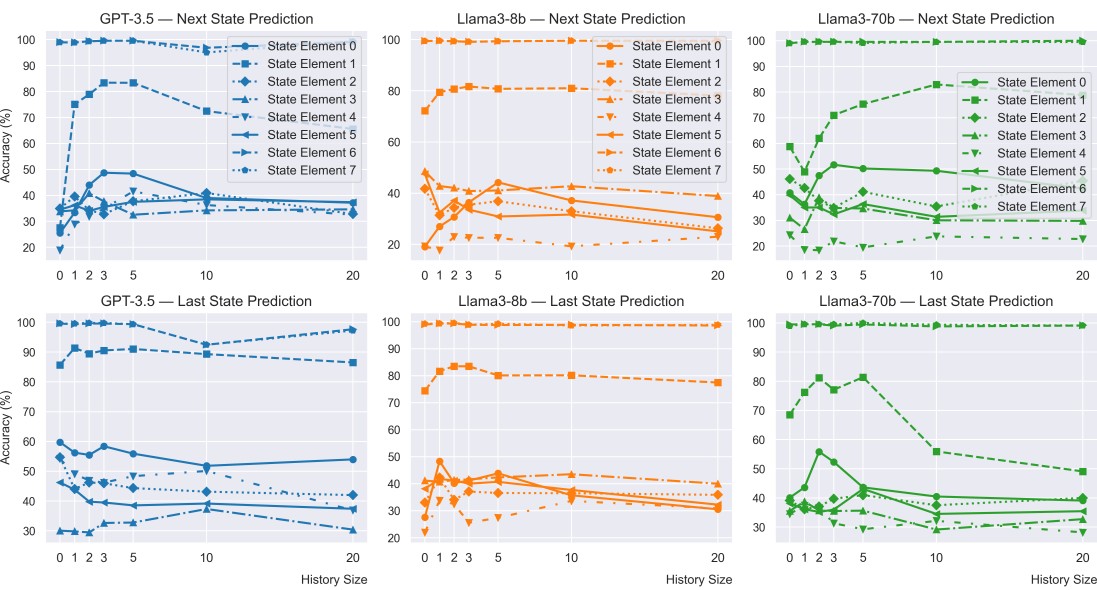

Figure 11: Dynamics of LLMs' performance on predicting individual state element for the **LunarLander** task (with indexed history in prompts).

## F.2 Average State Element Prediction Accuracy

In addition to reporting the dynamics of prediction accuracy for individual state elements, we report the averaged prediction accuracy for state elements in the MountainCar task (Figure 12), the Pendulum task (Figure 13), the Acrobot task (Figure 14), the LunarLander task (Figure 15), the FetchPickAndPlace task (Figure 16), the FetchPush task (Figure 17), and the FetchSlide task (Figure 18). Refer to Table 6 for the meanings of the state element indices in Fetch-series tasks.

We find that LLMs are slightly more sensitive to changes in *angular velocity* than *angle*, as shown by the Pendulum and Acrobot results. In addition, LLMs show greater sensitivity to changes in *position* than *linear velocity*, as demonstrated by MountainCar and LunarLander results.

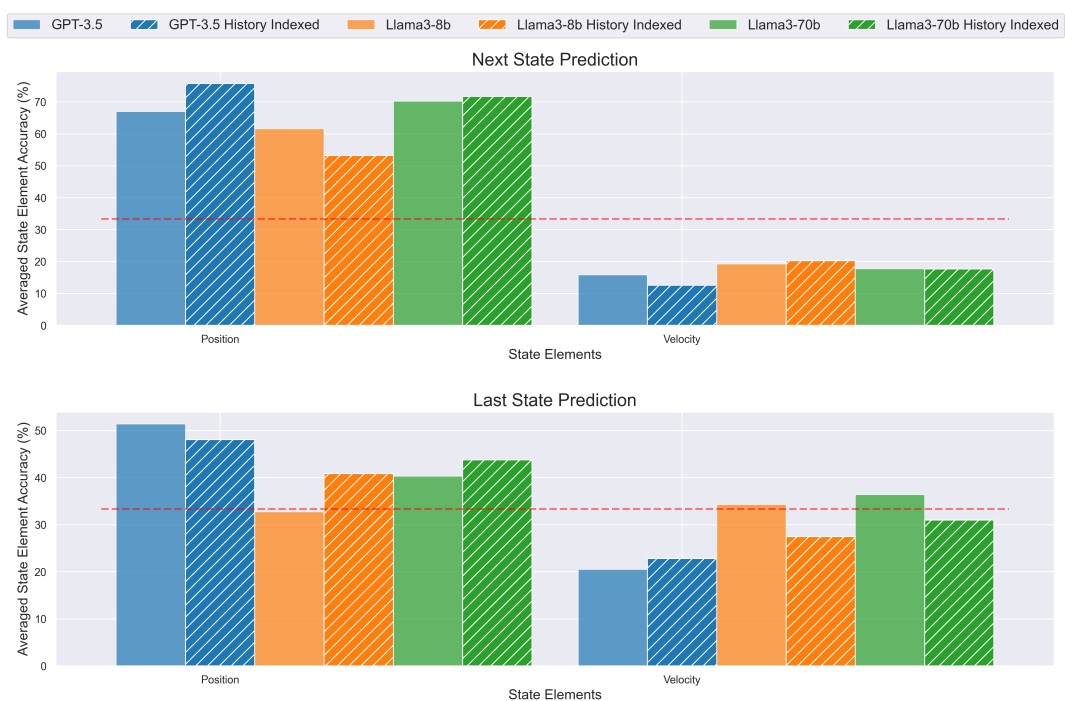

Figure 12: LLMs' averaged performance in predicting individual state element for the **MountainCar** task. The red dotted line represents the accuracy (33.33%) of randomly guessing state changes for each element (applicable to the following figures).

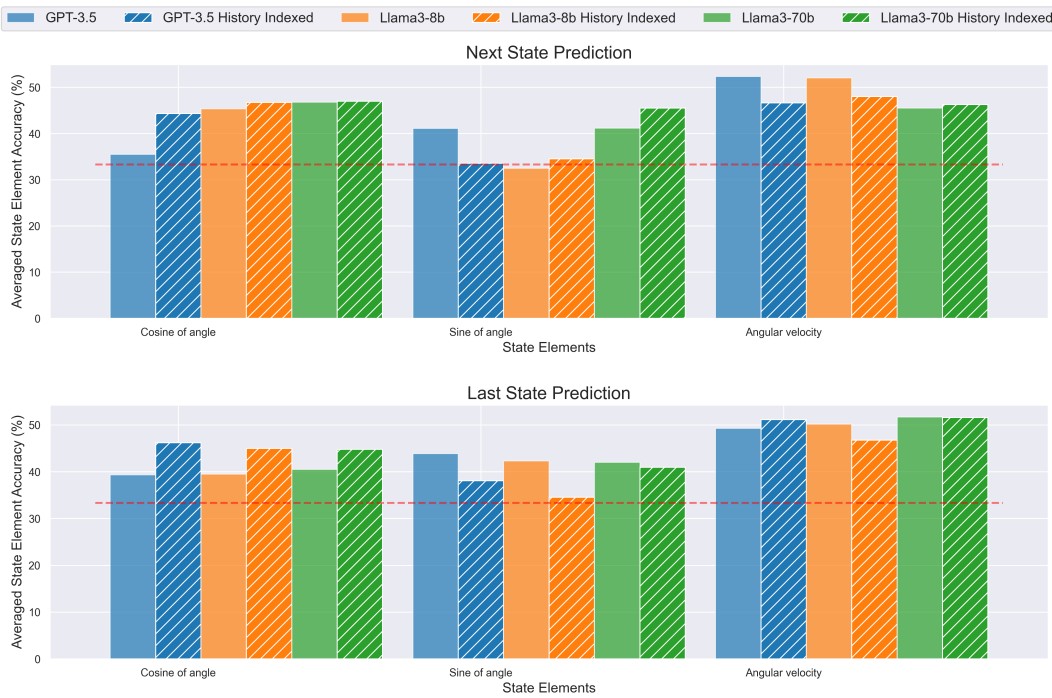

Figure 13: LLMs' averaged performance in predicting individual state element for the **Pendulum** task.

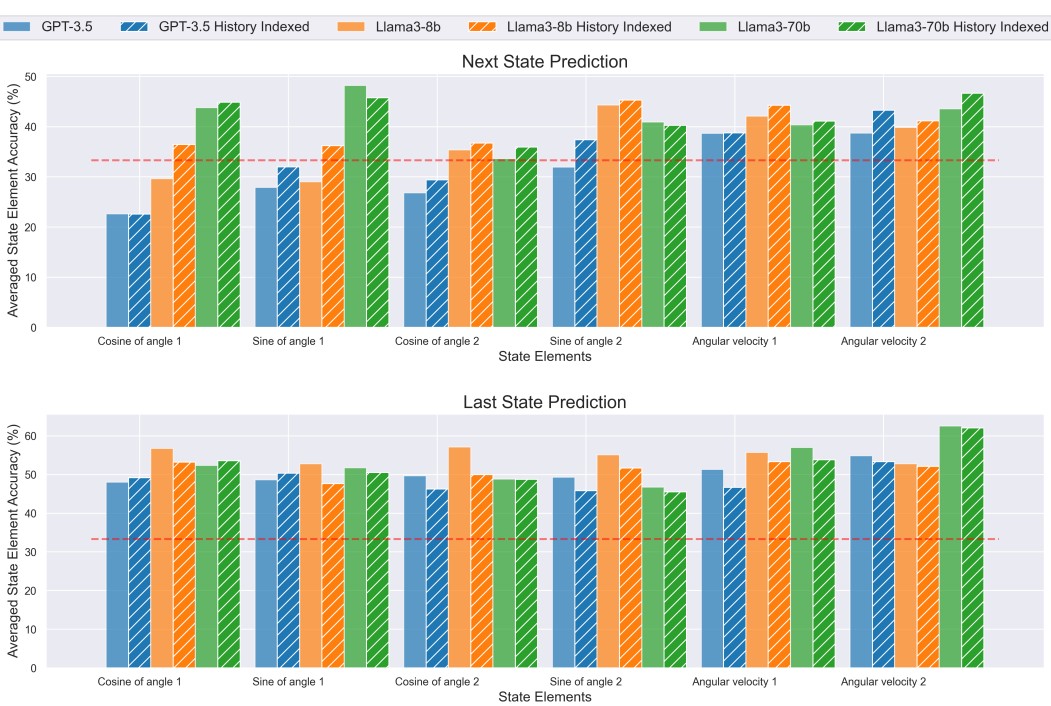

Figure 14: LLMs' averaged performance in predicting individual state element for the **Acrobot** task.

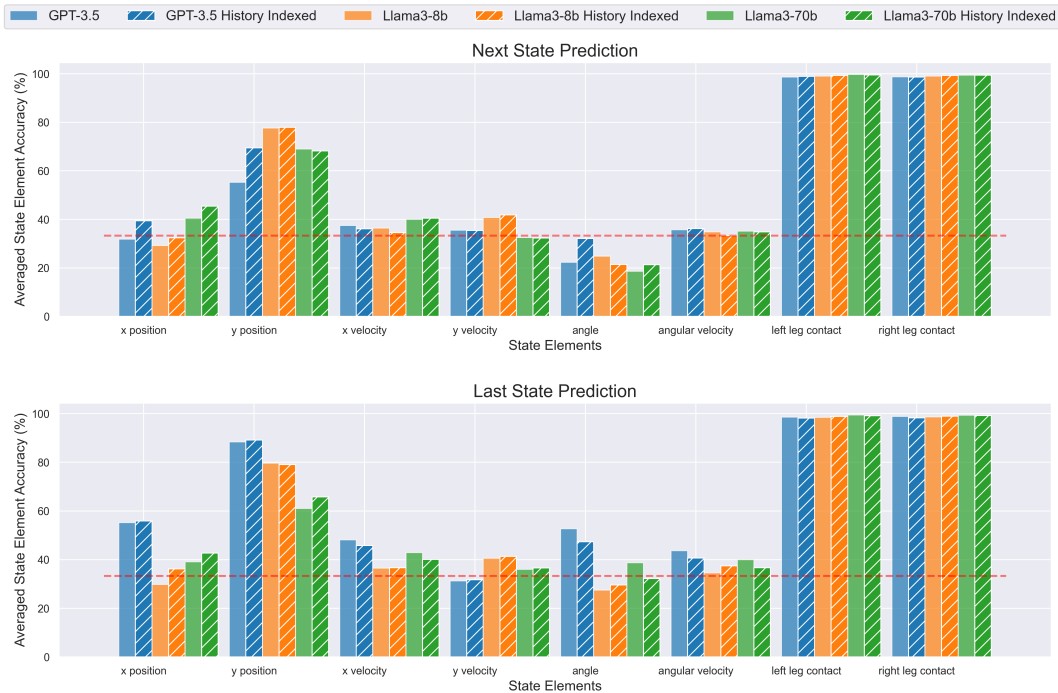

Figure 15: LLMs' averaged performance in predicting individual state element for the **LunarLander** task.

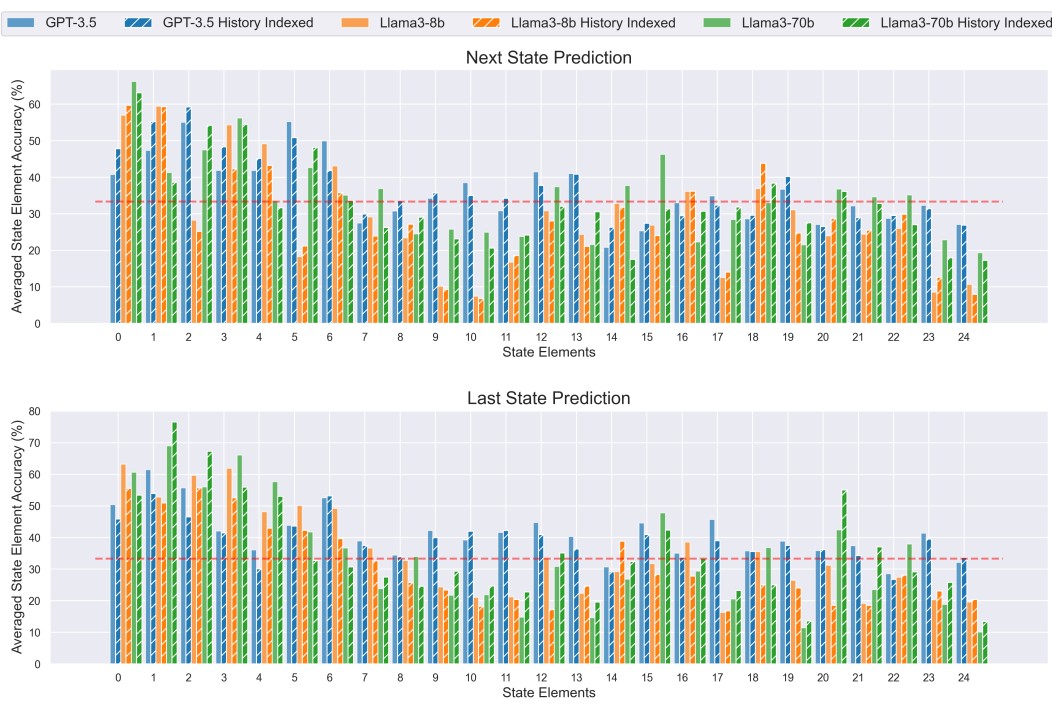

Figure 16: LLMs' averaged performance in predicting individual state element for the **FetchPickAndPlace** task. Refer to Table 6 for the meanings of the state element indices in Fetch-series tasks.

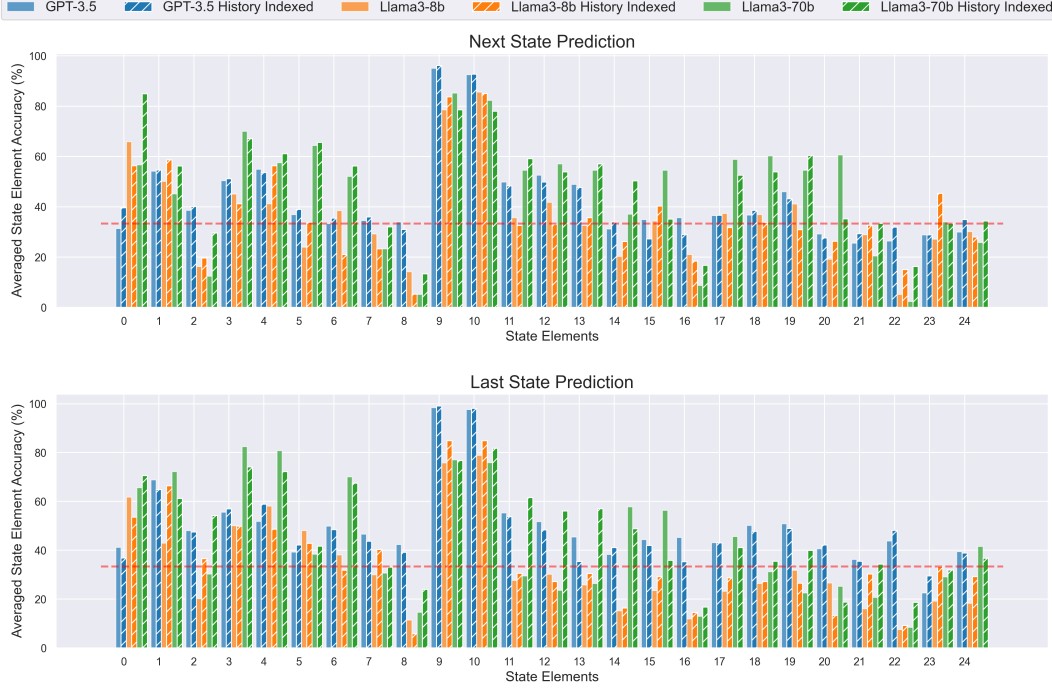

Figure 17: LLMs' averaged performance in predicting individual state element for the **FetchPush** task.

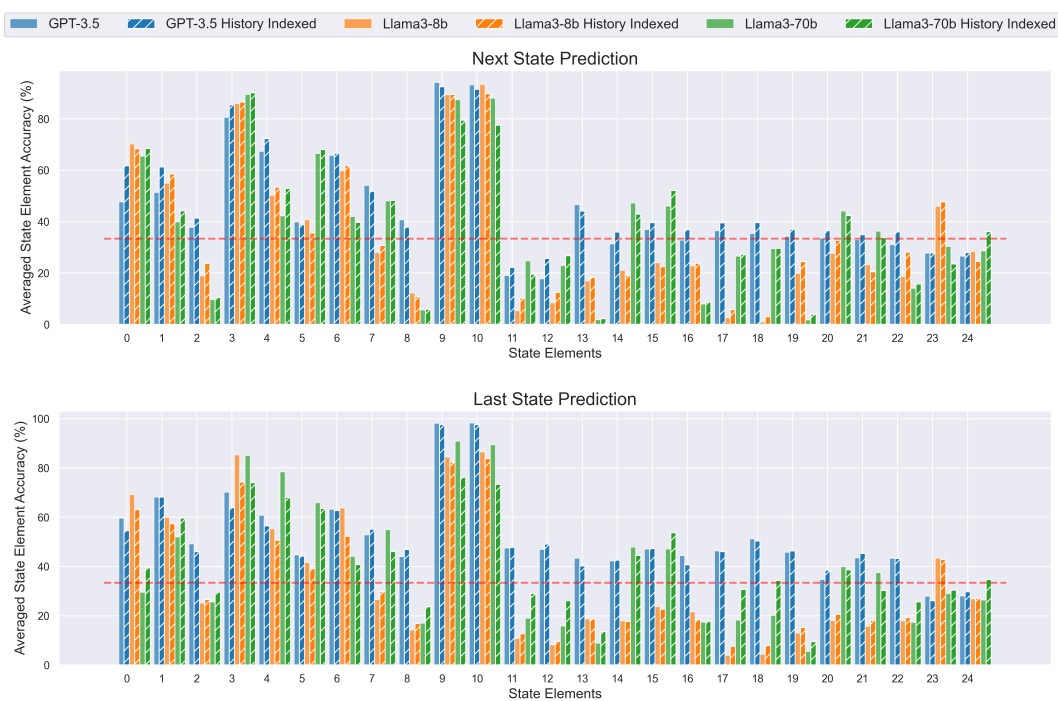

Figure 18: LLMs' averaged performance in predicting individual state element for the **FetchSlide** task.

Table 6: List of state elements in Fetch-series tasks

| Indices | State Elements Meanings |
|---------|-------------------------|
| 0 | End effector $x$ position |
| 1 | End effector $y$ position |
| 2 | End effector $z$ position |
| 3 | Block $x$ position |
| 4 | Block $y$ position |
| 5 | Block $z$ position |
| 6 | Relative block $x$ position with respect to gripper $x$ position |
| 7 | Relative block $y$ position with respect to gripper $y$ position |
| 8 | Relative block $z$ position with respect to gripper $z$ position |
| 9 | Joint displacement of the *right* gripper finger |
| 10 | Joint displacement of the *left* gripper finger |
| 11 | Global $x$ rotation of the block in a $XYZ$ Euler frame rotation |
| 12 | Global $y$ rotation of the block in a $XYZ$ Euler frame rotation |
| 13 | Global $z$ rotation of the block in a $XYZ$ Euler frame rotation |
| 14 | Relative block velocity in $x$ direction with respect to the gripper |
| 15 | Relative block velocity in $y$ direction with respect to the gripper |
| 16 | Relative block velocity in $z$ direction with respect to the gripper |
| 17 | Block angular velocity along the $x$ axis |
| 18 | Block angular velocity along the $y$ axis |
| 19 | Block angular velocity along the $z$ axis |
| 20 | End effector velocity in $x$ direction |
| 21 | End effector velocity in $y$ direction |
| 22 | End effector velocity in $z$ direction |
| 23 | *Right* gripper finger velocity |
| 24 | *Left* gripper finger velocity |

### F.3 Influential Factors in State Element Prediction

Figure 19 shows the linear relationship between state span and prediction accuracy, while Figure 20 shows the corresponding residuals-fitted plot. Figure 21 illustrates the relationship between state variance and prediction accuracy. Detailed state span (min-max) statistics for each state element across all tasks are detailed in Figure 22. For each task in Figure 19 and Figure 21, we used the raw state span during analysis, as LLMs process the unscaled state values. The linear analysis was conducted between the raw state span and prediction accuracy. However, to compare relationships across tasks, we used a 0-1 dummy scale for plotting purposes, as shown in Figure 7.

The linear relationship analysis for different models and tasks reveals key insights. For GPT-3.5 on the MountainCar task, the positive slope indicates that accuracy improves with span, and the perfect $r^2$ of 1 shows a flawless fit, meaning the model's predictions align perfectly with the data. For GPT-3.5 on the Acrobot task, the moderate slope and $r^2$ of 0.58 suggest a decent correlation with some unexplained variability. In contrast, the Llama3-70b model on the Pendulum task shows a low or near-zero slope and an $r^2$ of 0, indicating the model does not effectively explain the variability in the data for this task.

State element variance also influences dynamics prediction, being slightly more significant than state span in tasks like Pendulum and LunarLander, as shown in Figure 21. This offers a different perspective on how LLMs' dynamics understanding can be affected and generalized. Additionally, the linear relationship is often less significant for smaller models, suggesting that larger models are more sensitive to these changes. In general, from a statistical point of view, larger spans and greater state variance facilitate the LLMs' perception of changes due to larger value discrepancies.

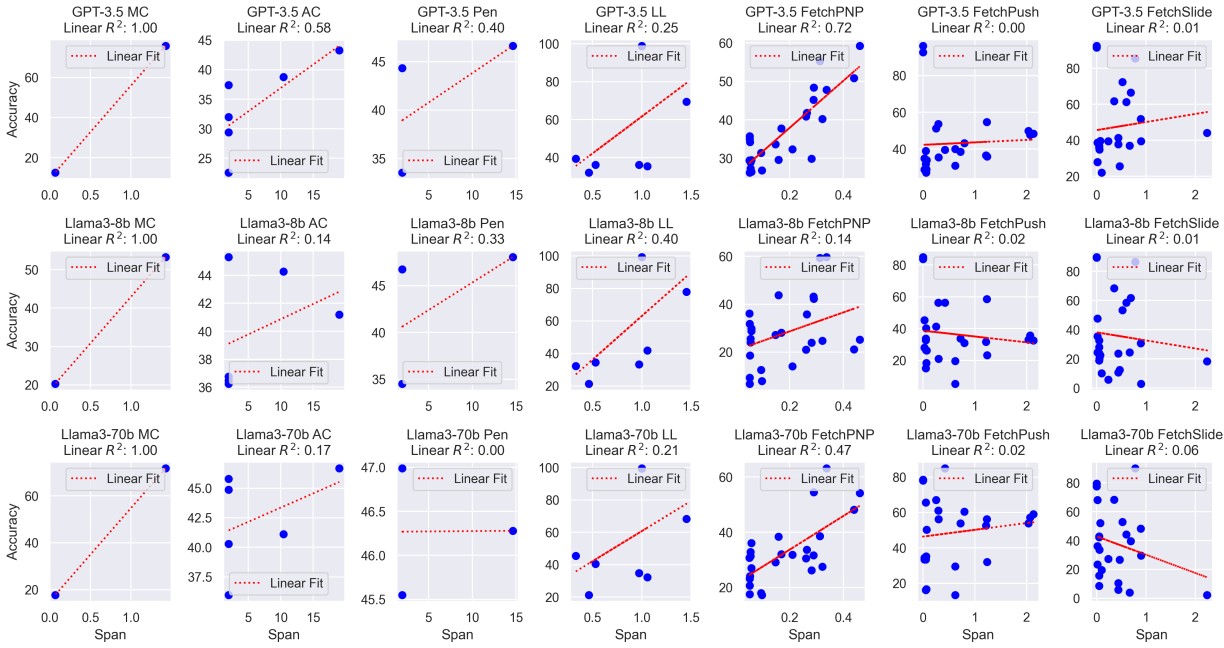

Figure 19: The linear relationship between *state span* and state element prediction accuracy across all LLMs and tasks. Each blue dot represents the average prediction accuracy for the corresponding state element. "MC" stands for MountainCar, "AC" for Acrobot, "Pen" for Pedulum, "LL" for LunarLander, "FetchPNP" for FetchPickAndPlace.

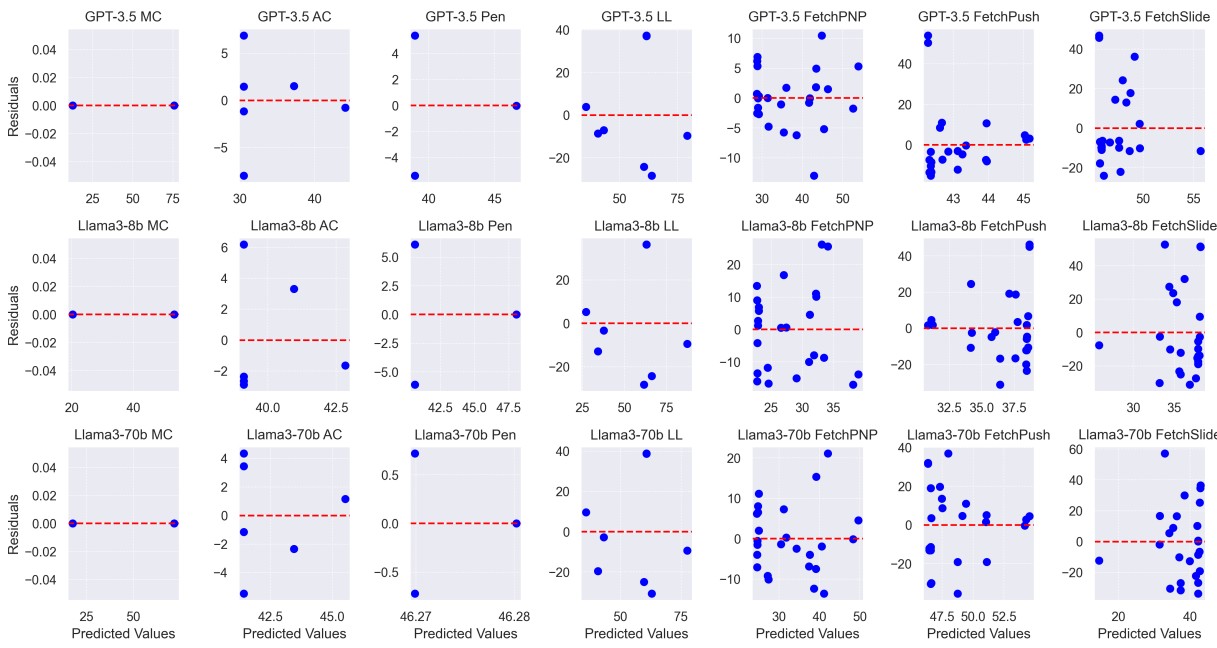

Figure 20: The residuals versus fitted values across all LLMs and tasks. Randomly distributed residuals around zero indicate that the linear models are appropriate for the task data.

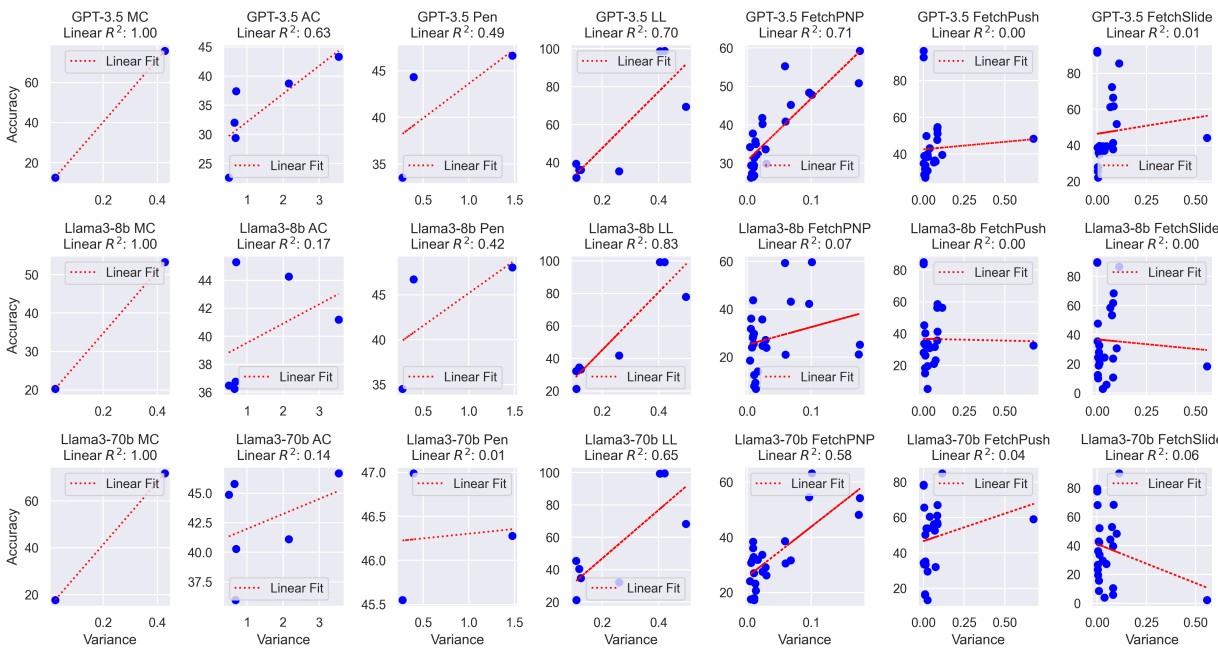

Figure 21: The linear relationship between *state variance* and state element prediction accuracy across all LLMs and tasks.

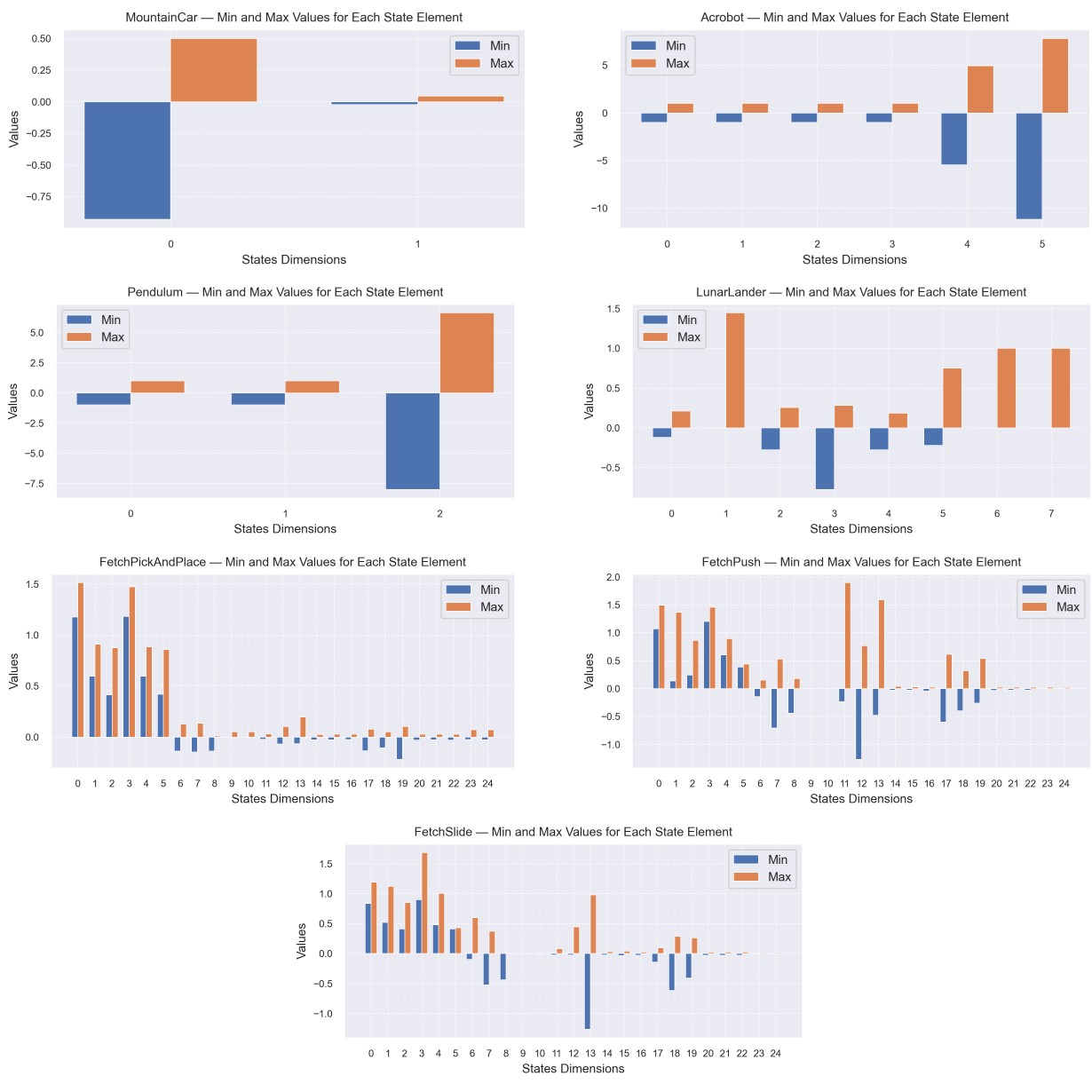

Figure 22: Minimum and maximum values of state elements across all tasks.

### F.4 Average Action Element Prediction Accuracy

It is observed that all LLMs fall short of scoring reasonable accuracy in predicting 4-dim actions for the Fetch-series tasks compared to the four classic control tasks. Therefore we report the averaged prediction accuracy for individual action elements in the FetchPickAndPlace (Figure 23), the FetchPush (Figure 24), and the FetchSlide tasks (Figure 25).

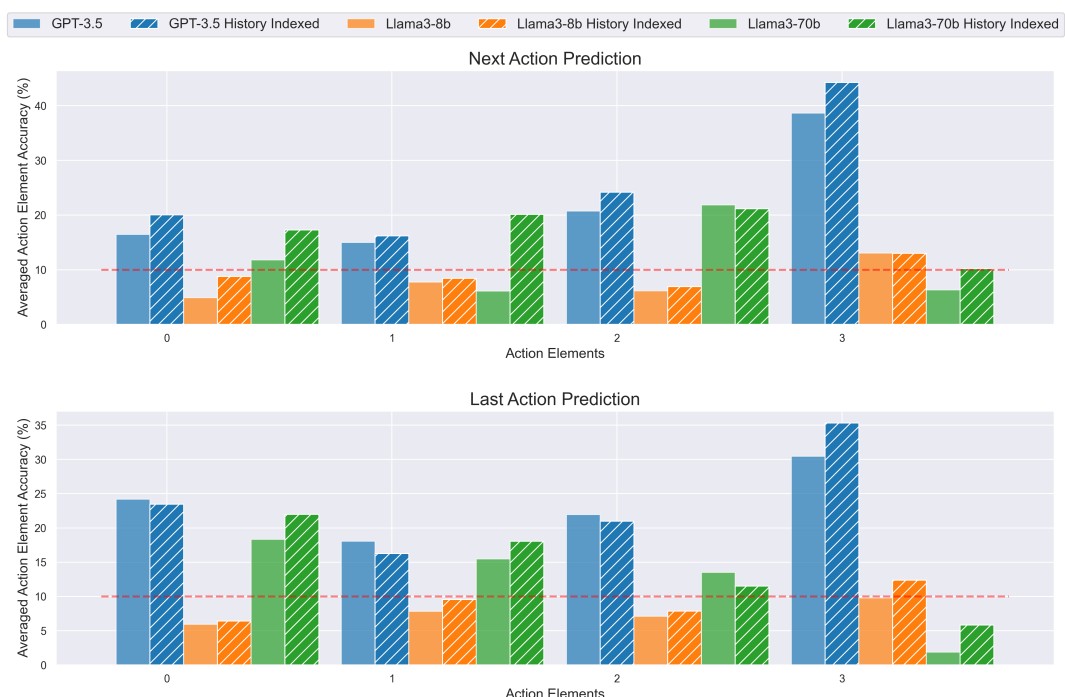

Figure 23: LLMs' averaged performance in predicting individual action element for the **FetchPickAndPlace** task. "0" represents $x$ displacement, "1" $y$ displacement, and "2" $z$ displacement, "3" gripper displacement. The red dotted line represents the accuracy (10%) of randomly guessing action bins (total of 10) for each element, applicable to the following figures.

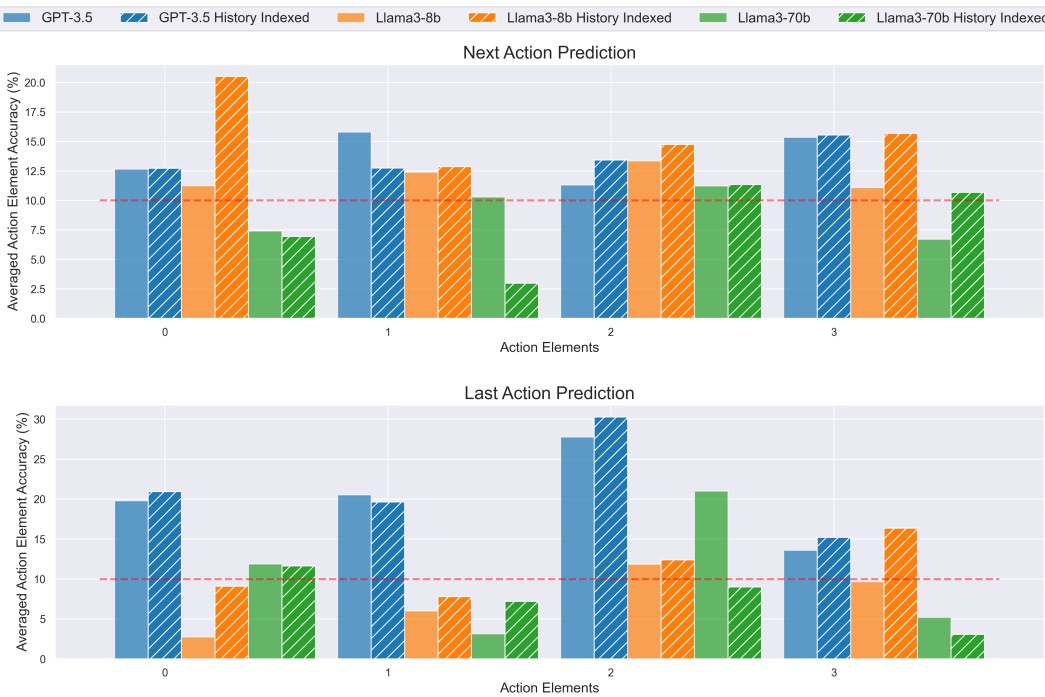

Figure 24: LLMs' averaged performance in predicting individual action element for the **FetchPush** task. "0" represents $x$ displacement, "1" $y$ displacement, and "2" $z$ displacement, "3" gripper displacement.

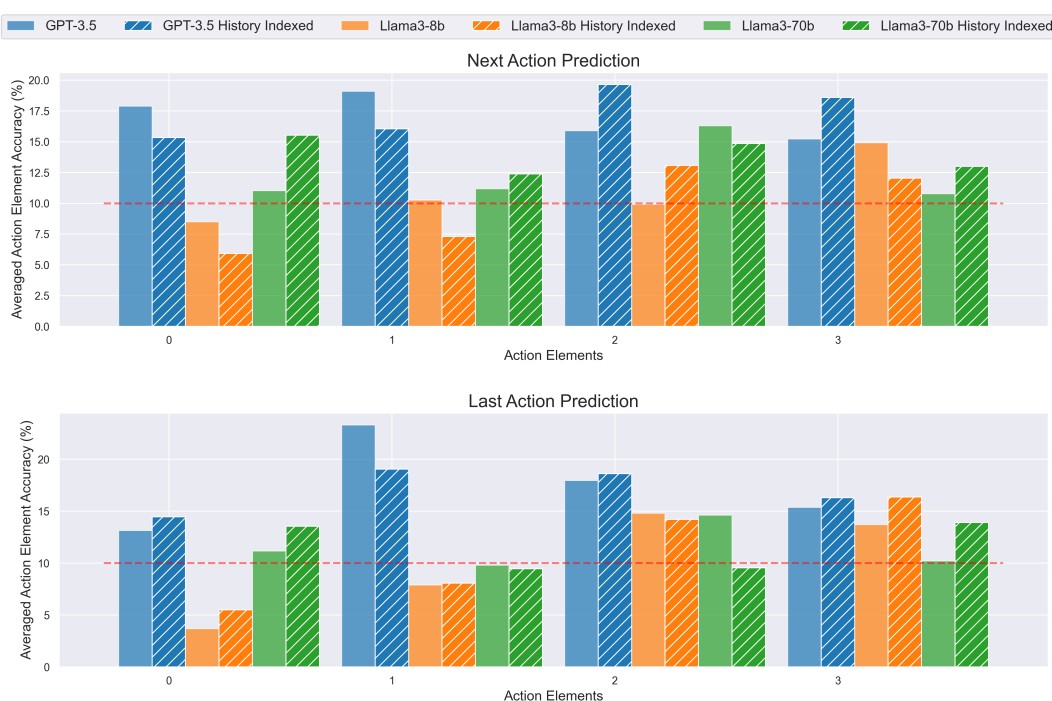

Figure 25: LLMs' averaged performance in predicting individual action element for the **FetchSlide** task. "0" represents $x$ displacement, "1" $y$ displacement, and "2" $z$ displacement, "3" gripper displacement.

## F.5   Average Comparison of Model Predictions

Table 7 displays the average accuracy of LLMs' predictions regarding the agent's behaviour and the resulting state changes in Mountaincar, Acrobot, and Pendulum tasks.

| | MountainCar | | | Acrobot | | | Pendulum | | |
|---|---|---|---|---|---|---|---|---|---|
| | **GPT-3.5** | **Llama3-8b** | **Llama3-70b** | **GPT-3.5** | **Llama3-8b** | **Llama3-70b** | **GPT-3.5** | **Llama3-8b** | **Llama3-70b** |
| **NA Pred.** | 74.60% | 59.10% | **86.18%** | 43.94% | 46.29% | **65.12%** | **17.08%** | 3.72% | 11.77% |
| | 81.48% ↑ | 68.63% ↑ | **87.06%** ↑ | 46.36% ↑ | 44.95% ↓ | **64.73%** ↓ | **17.49%** ↑ | 3.51% ↓ | 12.42% ↑ |
| **LA Pred.** | 76.73% | 61.83% | **78.87%** | 39.25% | 47.24% | **55.32%** | **14.28%** | 1.58% | 14.02% |
| | **80.06%** ↑ | 73.99% ↑ | 76.85% ↓ | 44.62% ↑ | 42.35% ↓ | **55.40%** ↑ | **20.63%** ↑ | 1.89% ↑ | 13.86% ↓ |
| **NS Pred.** | 33.43% | 30.81% | **37.04%** | **0.30%** | 0.26% | 0.13% | **9.52%** | 8.34% | 7.61% |
| | 37.41% ↑ | 33.65% ↑ | **40.68%** ↑ | 0.00% ↓ | 0.42% ↑ | **0.43%** ↑ | 7.89% ↓ | 6.65% ↓ | 5.49% ↓ |
| **LS Pred.** | **31.97%** | 22.12% | 29.32% | 1.14% | **2.95%** | 1.69% | 6.46% | **10.54%** | 10.22% |
| | 32.41% ↑ | 22.45% ↑ | **35.25%** ↑ | 0.61% ↓ | 2.32% ↓ | **2.87%** ↑ | 5.41% ↓ | **8.27%** ↓ | 7.64% ↓ |

Table 7: Comparison of model predictions with and w/o indexed history. Light grey cells show results with **indexed history**. NA Pred. = Next Action Prediction; LA Pred. = Last Action Prediction; NS Pred. = Next State Prediction; LS Pred. = Last State Prediction.

## F.6   Dynamic Performance of All Evaluation Metrics

The dynamics of LLMs' understanding performance (with and without indexed history in prompts) with increasing history size for the MountainCar task (Figure 26), the Acrobot task (Figure 27), the Pendulum task (Figure 28 and Figure 29), and the LunarLander task (Figure 30).

Among all results, it is observed that models' understanding of agent behaviour improves significantly with smaller history sizes but does not increase further with larger histories. In some cases, like with Llama3-70b,

it may even degrade. Generally, model performance in action prediction tends to increase and then likely saturate as the history size grows. The optimal history size for each language model is detailed in Table 8.

In complex tasks like Acrobot, history size has less impact on model performance in state prediction. We hypothesise that this is due to the complex relationships in the interaction data, where adding more history does not enhance the LLMs' understanding of the environment dynamics. For moderately complex tasks (e.g., Pendulum), model performance initially increases with a small history size, consistent with our earlier finding for predicting actions. This is demonstrated in the third column of Figure 28.

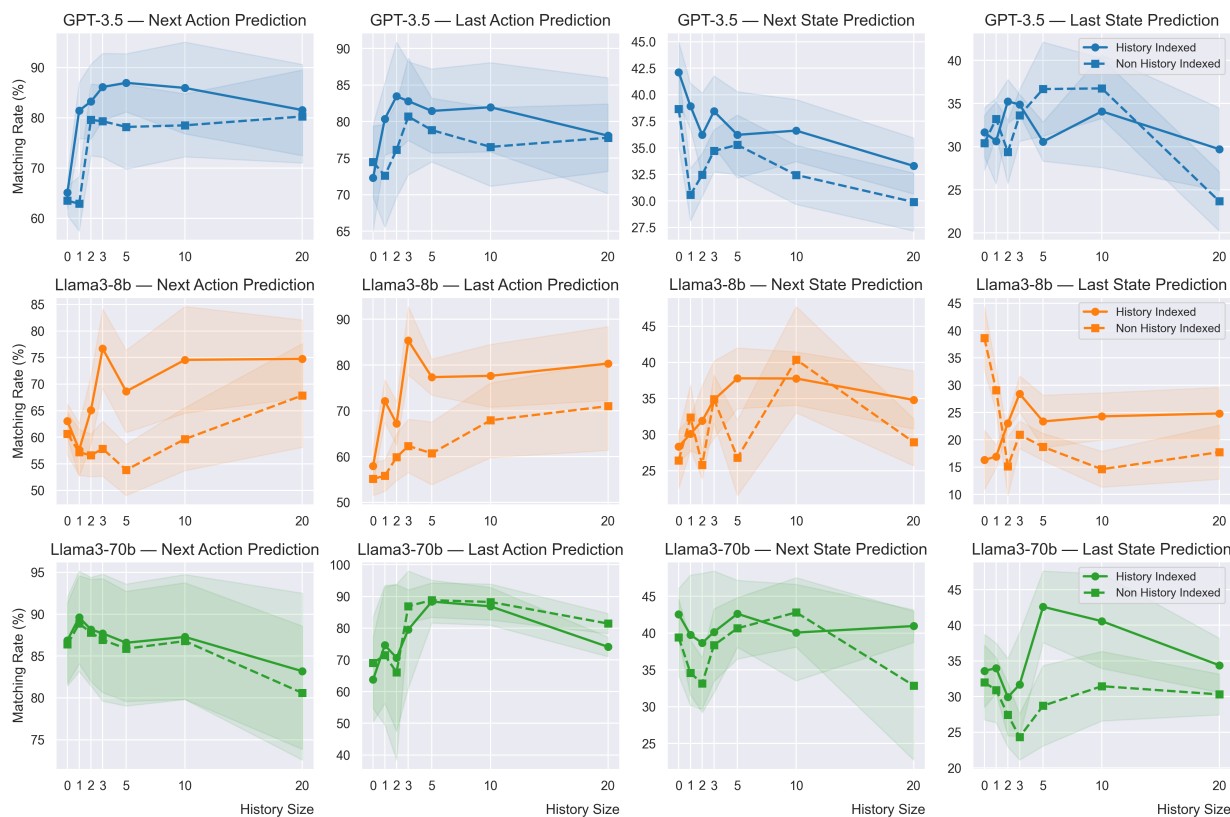

Figure 26: The dynamics of LLMs' understanding performance with increasing history size for the **MountainCar** task.

| Model | MountainCar | Acrobot | Pendulum | LunarLander |
|:---:|:---:|:---:|:---:|:---:|
| GPT-3.5 | 3 | 5 | 5 | 5 |
| Llama3-8b | 3 | 20 | 1 | 5 |
| Llama3-70b | 5 | 0 | 1 | 10 |
| GPT-4o | 1 | 1 | 2 | 1 |

Table 8: Optimal history size for each language model, evaluated across four metrics for individual tasks.

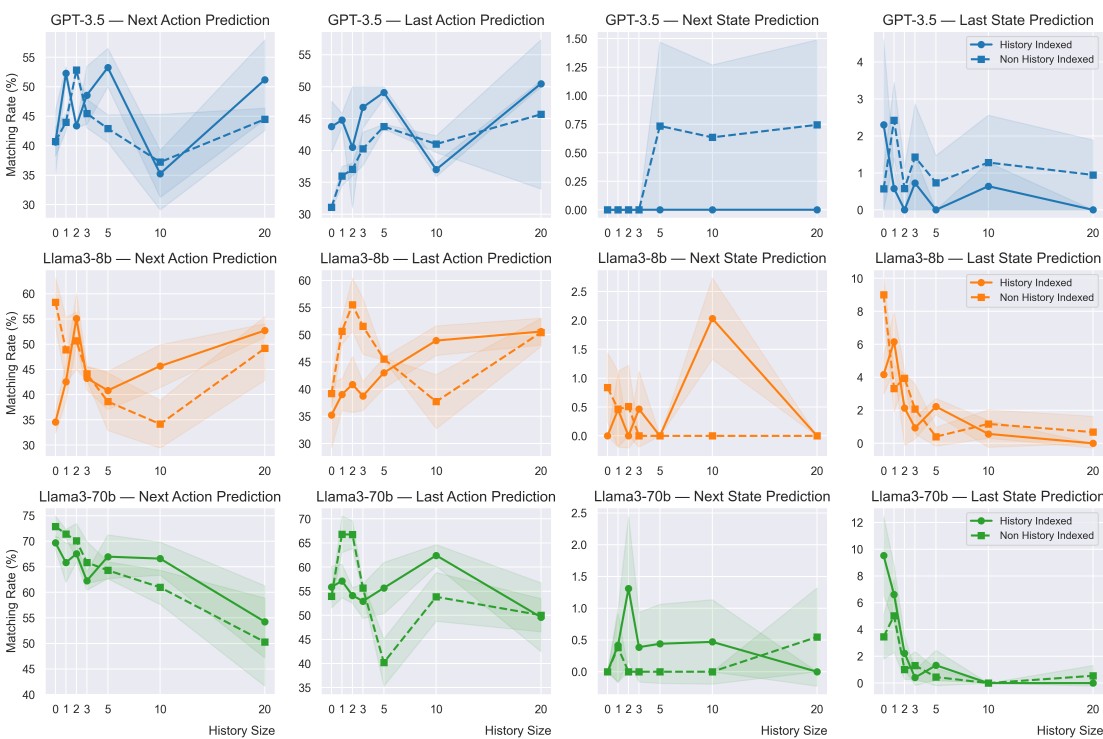

Figure 27: The dynamics of LLMs' understanding performance with increasing history size for the **Acrobot** task.

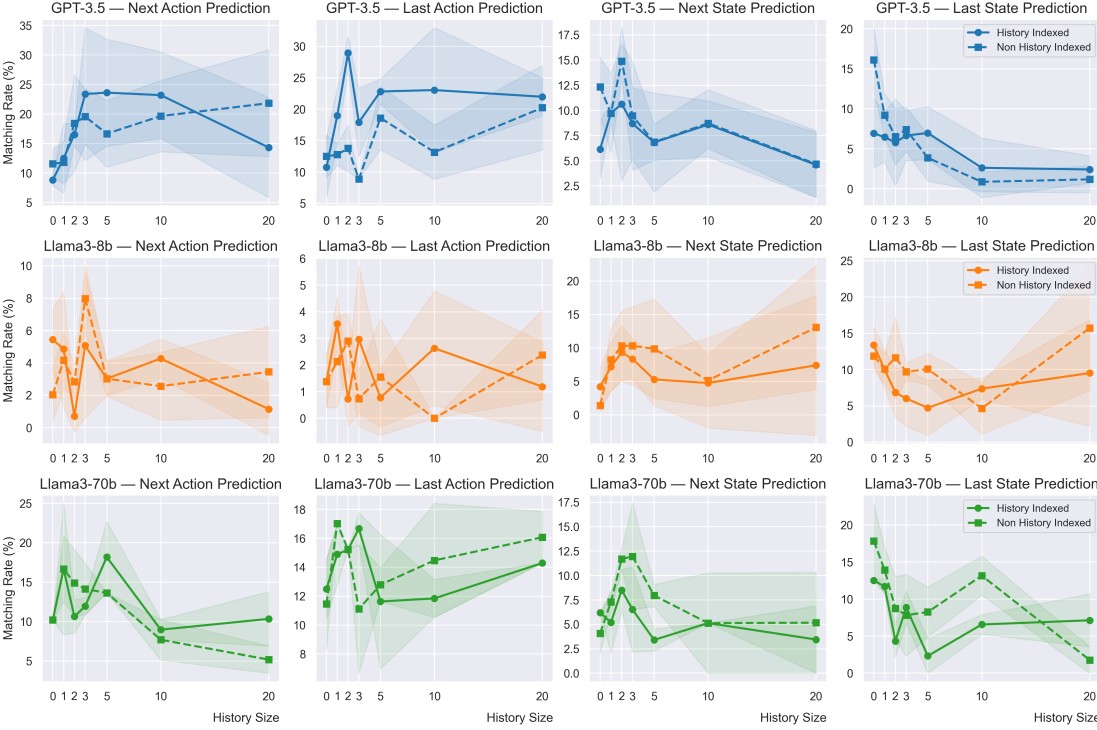

Figure 28: The dynamics of LLMs' understanding performance with increasing history size for the **Pendulum** task with **discretized** actions in evaluation prompts.

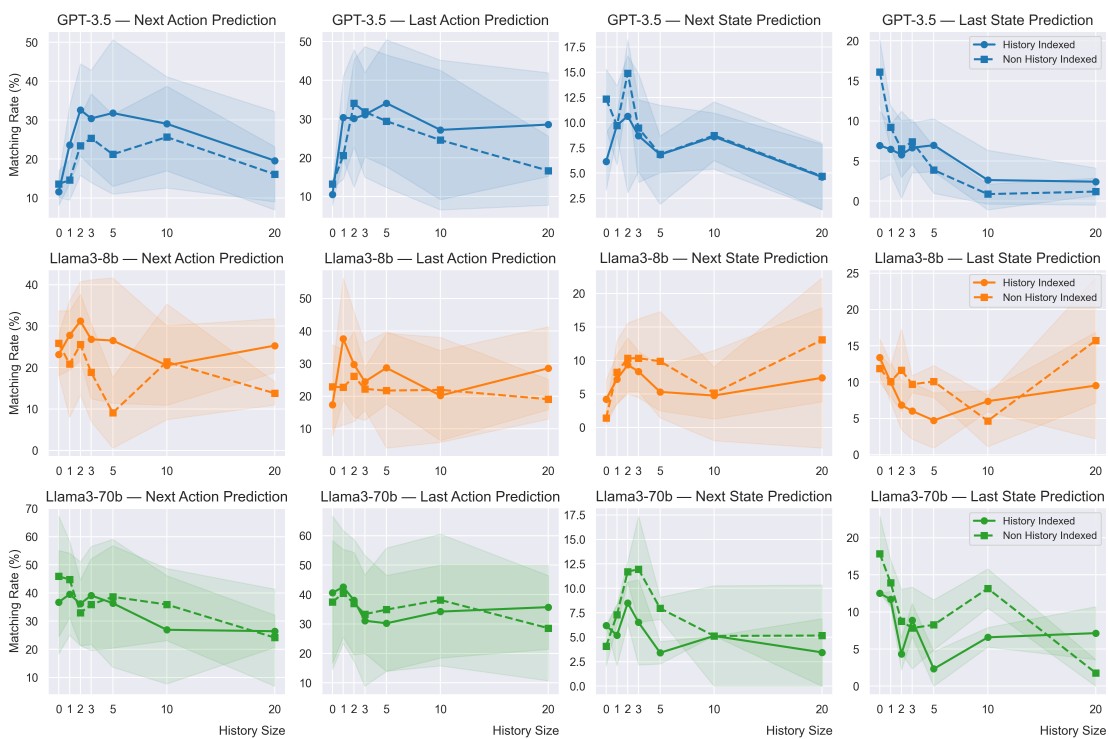

Figure 29: The dynamics of LLMs' understanding performance with increasing history size for the **Pendulum** task with **continuous** actions in evaluation prompts.

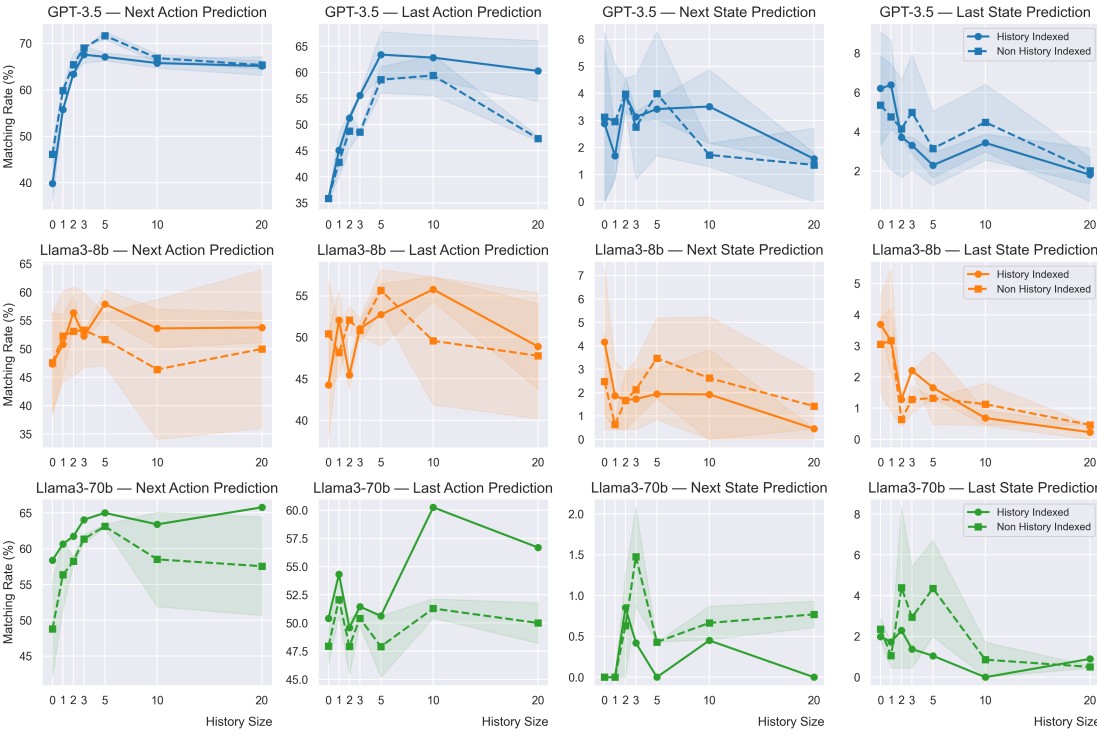

Figure 30: The dynamics of LLMs' understanding performance with increasing history size for the **LunarLander** task.

### F.6.1 Comparative performance of models on predicting discretized actions

Continuing from the plot of LLMs' performance on the Pendulum task with continuous actions (third row of Figure 4 in the main text), Figure 31 presents a comparative plot of LLMs' performance on the Pendulum task with **discretized** actions. The results show that LLMs are generally less effective at classifying action bins compared to regressing on absolute action values.

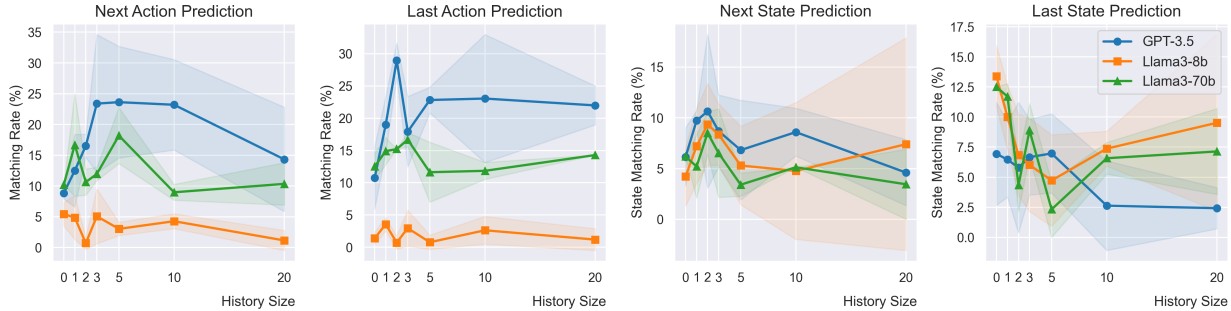

Figure 31: A comparative plot of LLMs' performance on the **Pendulum** task with **discretized** actions, following the plot of predicting continuous actions (third row of Figure 4 in the main text).

### F.6.2 Comparative performance of models on Fetch-series tasks

Continuing from Figure 4 in the main text, Figure 32 presents the performance of LLMs on the Fetch tasks, characterized by continuous actions ($d_{\mathcal{A}} = 4$) and a high-dimensional state space ($d_{\mathcal{S}} = 25$). The results show that LLMs struggle with state predictions, achieving zero accuracy with very large state elements.

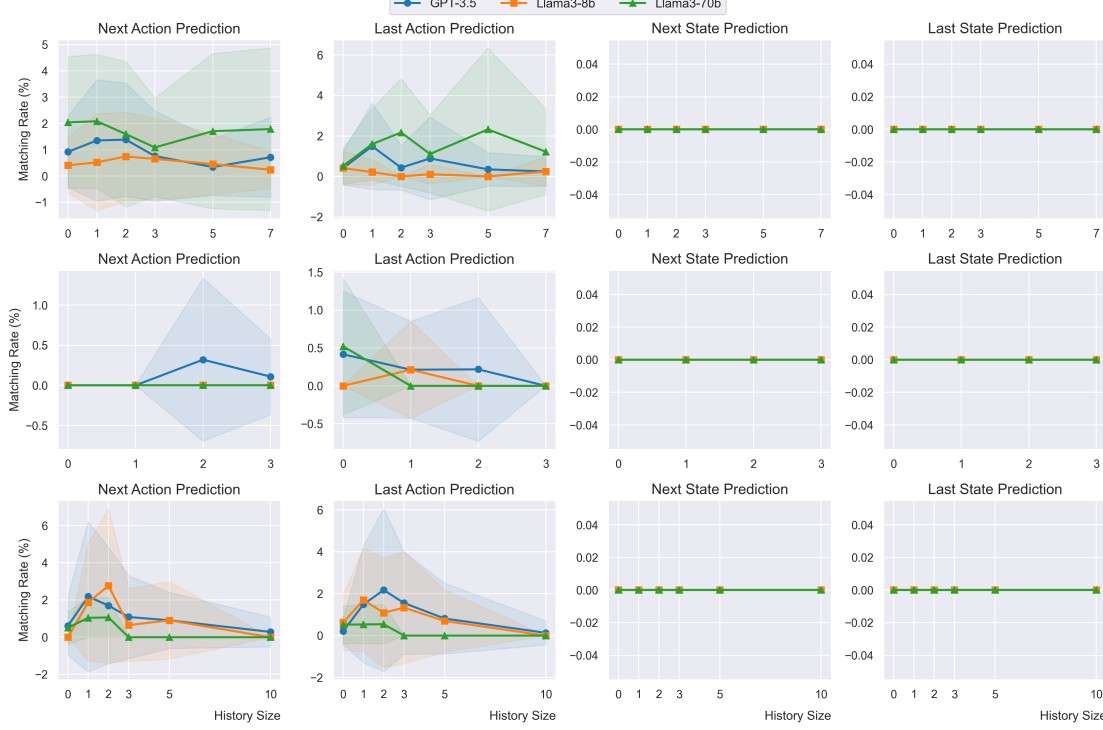

Figure 32: Comparative plots of LLMs' performance on the **Fetch-series** task, following the plots in Figure 4 in the main text).

### F.7 Relative Improvement and Worsening Rates with Increased History Size

When increasing history size in prompts for LLMs, we observed that previously incorrect predictions at certain time steps may become correct, and vice versa (see Figure 33 for an example with the MountainCar task). We define the relative improvement rate for a history size $H > 0$ compared to a fixed history size 0 as $\frac{\#(\text{incorrect}\rightarrow\text{correct})_{0\rightarrow H}}{\#\text{incorrect}_0}$ and the relative worsening rate for the same $H$ as $\frac{\#(\text{correct}\rightarrow\text{incorrect})_{0\rightarrow H}}{\#\text{correct}_0}$.

As shown in Figure 33, increasing history size does not impact steps near the episode end, where RL agents consistently accelerate to the right to cross the finish line. In contrast, LLMs, likely influenced by pre-training on human preferences, suggest more cautious actions, such as accelerating left to avoid overshooting. This safety bias, ingrained during pre-training, significantly affects their action predictions regardless of history length. Thus, LLM prediction failures are not solely due to difficulty processing extensive histories; their inherent beliefs about agent behaviour and environment context can also impair prediction accuracy in certain tasks.

Further, by examining step-wise mismatches in the Acrobot task, it is observed that LLMs often justify their predictions mainly on the rewards received in previous steps, neglecting history patterns, especially when the history is short.

Figure 34 and Figure 35 illustrate how these improvement/worsening rates change as history size increases. Analysing the impact of longer history on correcting previous incorrect predictions when $H$ is fixed at 0 shows that increasing history size improves LLMs' understanding of agent behaviour initially, but further increases may degrade it. Conversely, increasing history size has a negligible influence on dynamics predictions.

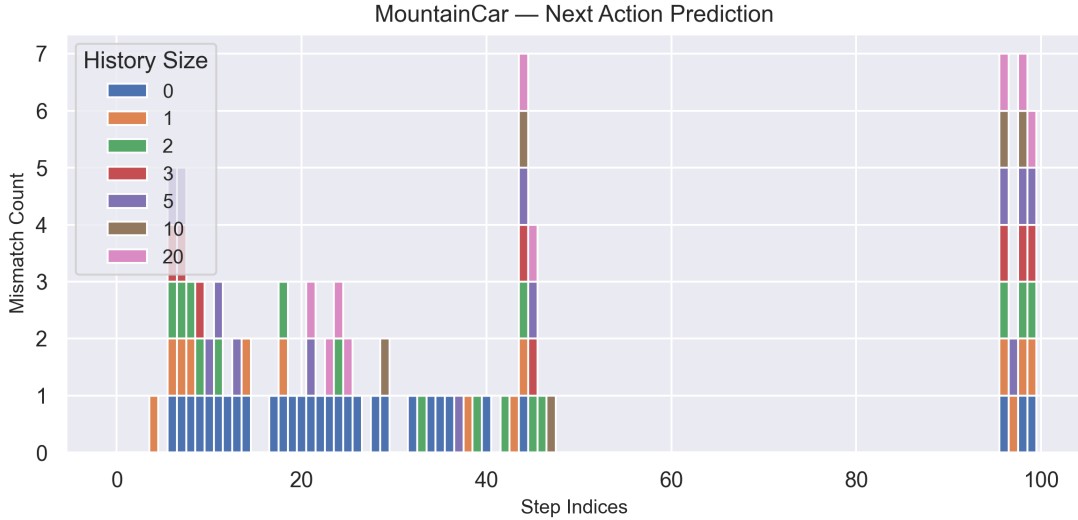

Figure 33: Step-wise mismatches with increasing history size for the GPT-3.5 model on the MountainCar task (the same mismatch pattern is observed across different episodes). Each bar (corresponding to a different history size) at each step index indicates a mismatch between the LLMs' prediction (based on that history size) and the action taken by the RL agent at that step.
At certain steps (e.g., around 40), the model with $h = 0$ fails, while the model with $h > 1$ succeeds. In contrast, at other steps (e.g., at the last steps), increasing history size minimally affects the LLMs' predictions.

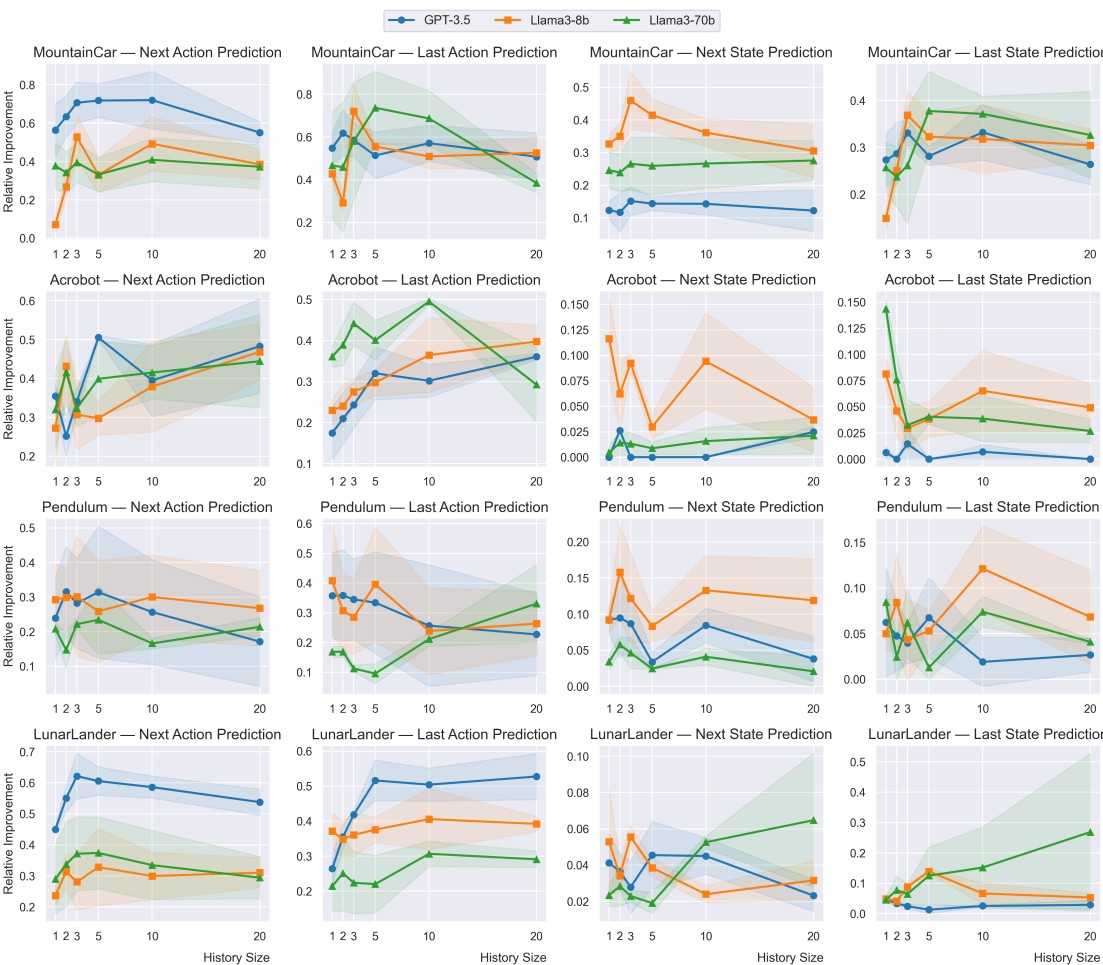

Figure 34: The dynamics of relative improvement with increasing history size across all LLMs and tasks (with *indexed history* in prompts). The first two columns represent *agent understanding*; the last two columns represent *dynamics understanding*.

## F.8 Ablation Study

### F.8.1 Comparison of models without using task dynamics

Figure 36 illustrates the performance variation when dynamics equations are excluded from the prompts.

### F.8.2 Comparison of models without using task instructions

Akin to prior works by Mishra et al. (2022); Le Scao & Rush (2021), which show that task framing in prompt influences language models, we observe a similar effect. When removing task instruction from evaluation prompts, models' understanding performance across the majority of evaluation metrics is significantly degrading, as demonstrated in MountainCar (Figure 37) and Acrobot (Figure 38) tasks; despite the history context (i.e., sequence of numerical values) remaining unchanged. We hypothesise that LLMs' ability to mental model agents is enhanced by a more informative context.

### F.8.3 Comparison of models: action bins vs. absolute values prediction

Figure 39 presents the evaluation results of LLMs on Pendulum tasks, comparing predictions of action bins (the first three rows) with predictions of absolute action values (the last three rows).

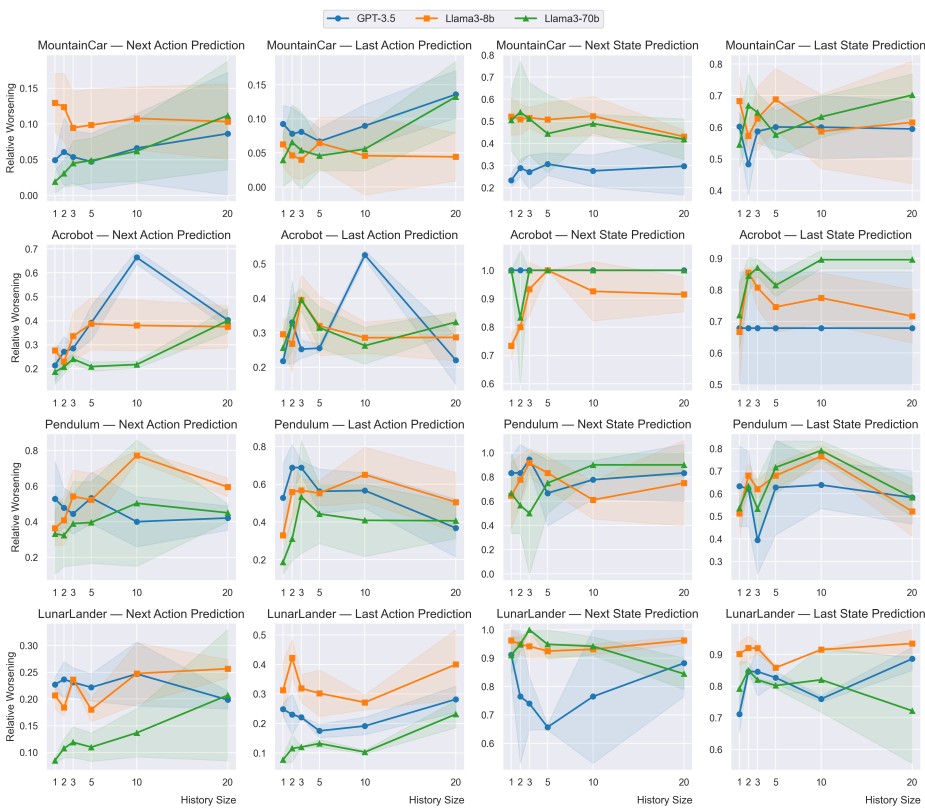

Figure 35: The dynamics of relative worsening with increasing history size across all LLMs and tasks (with *indexed history* in prompts).

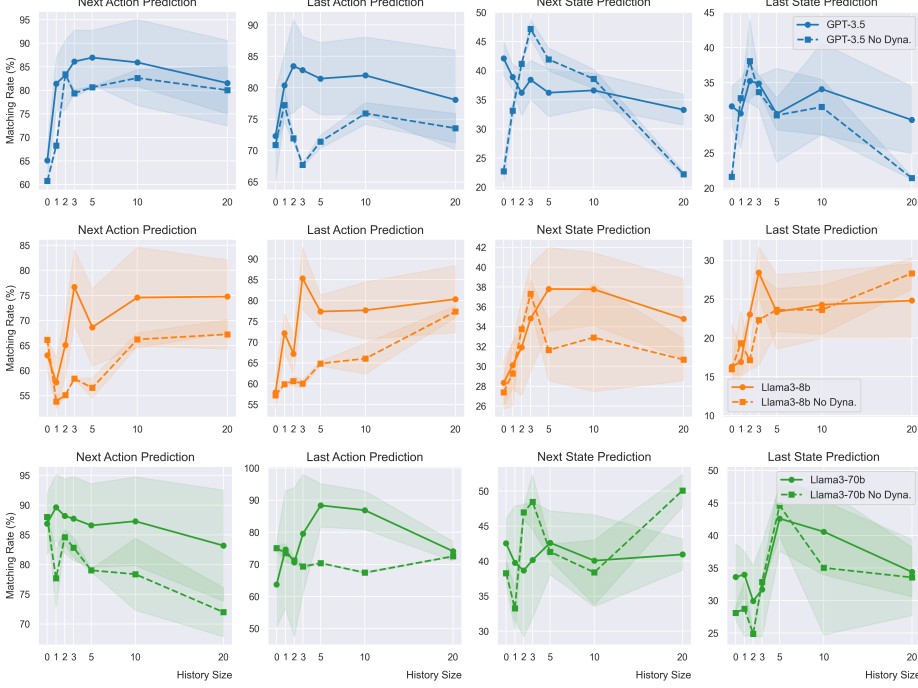

Figure 36: Comparative plots of LLMs' performance on **MountainCar** with different history sizes (with *indexed history* in prompts). The suffix of model names "No Dyna." indicates **not using dynamics equations** in prompts.

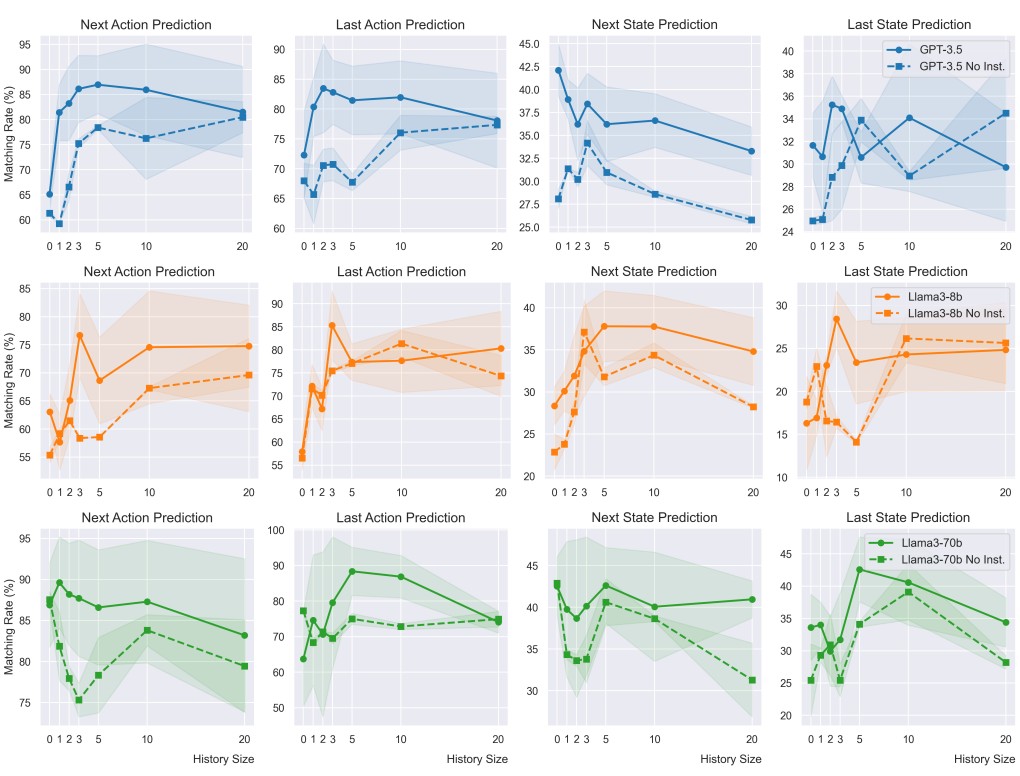

Figure 37: Comparative plots of LLMs' performance on **MountainCar** with different history sizes (with *indexed history* in prompts). The suffix of model names "No Inst." indicates **not using task description** in prompts.

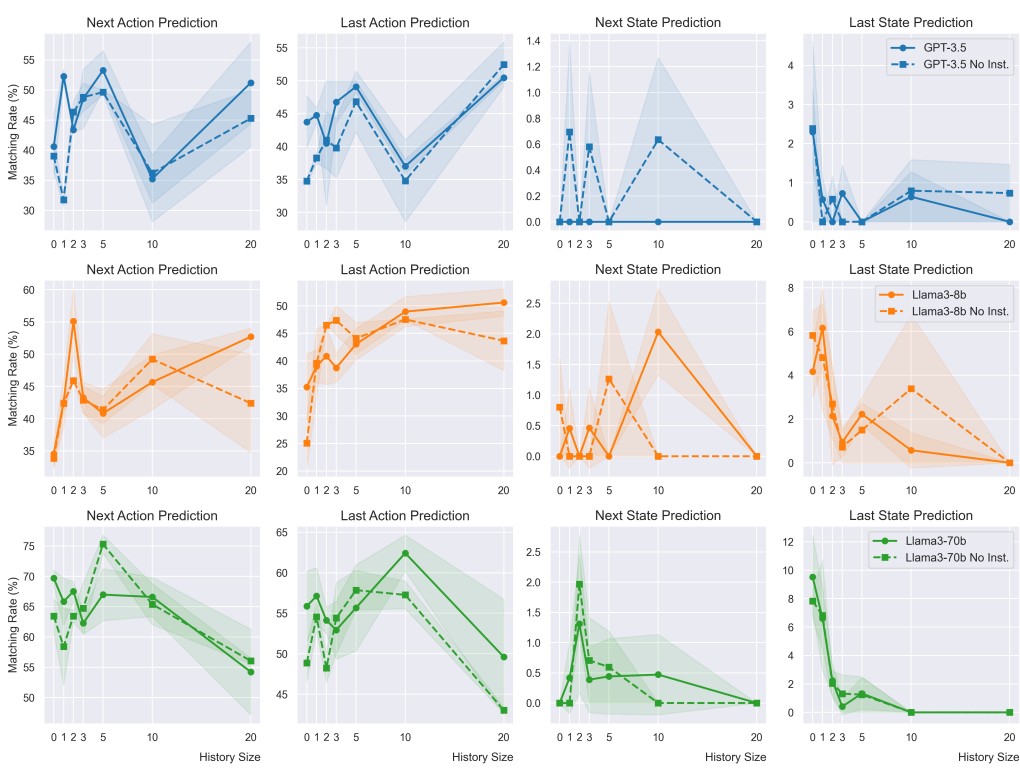

Figure 38: Comparative plots of LLMs' performance on **Acrobot** with different history sizes (with *indexed history* in prompts). The suffix of model names "No Inst." indicates **not using task description** in prompts.

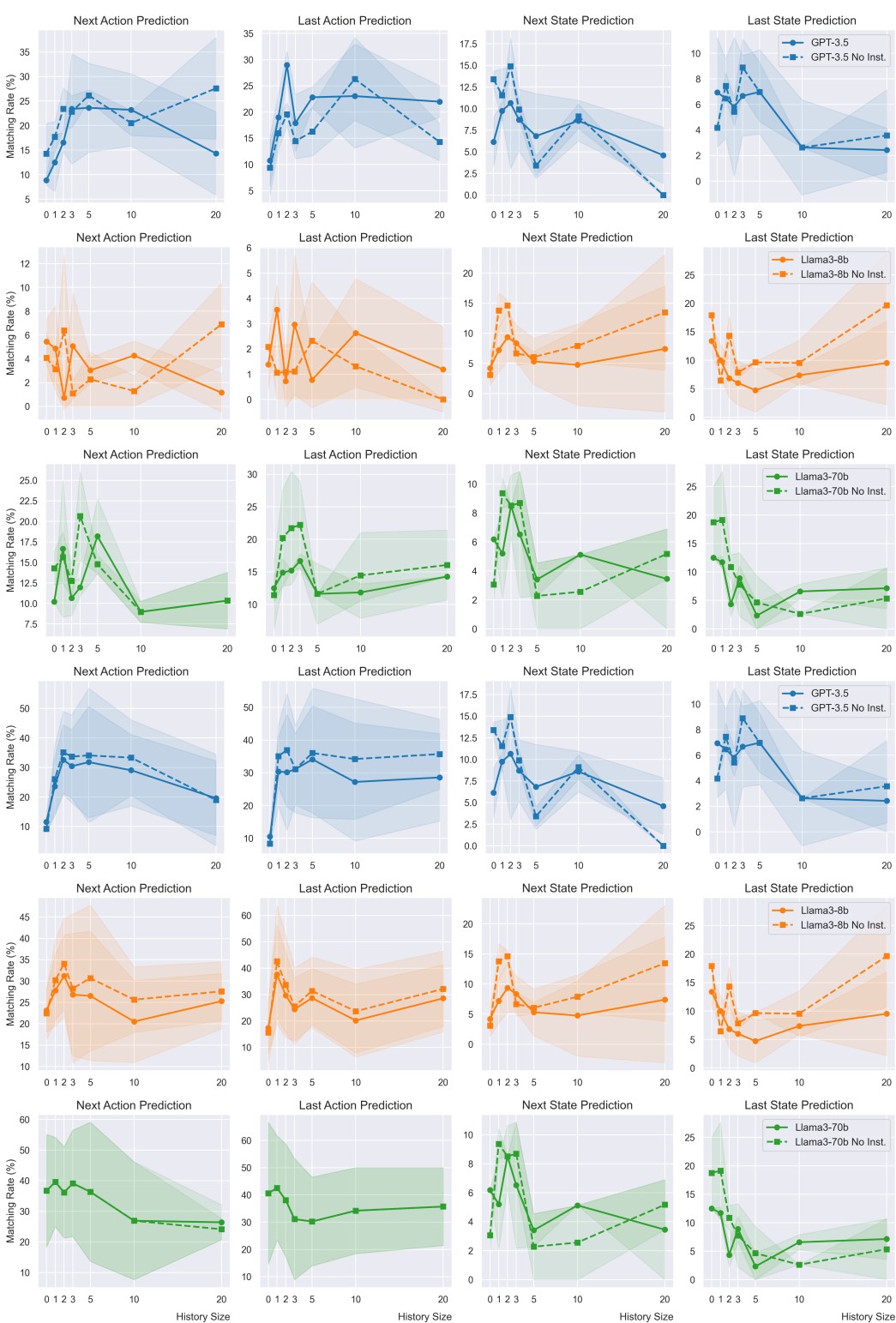

Figure 39: Comparative plots of LLMs' performance on **Pendulum** with different history sizes (with *indexed history* in prompts). First three rows: *predicting action bins*; Last three rows: *predicting absolute action values.*

## G  LLMs Erroneous Responses in MountainCar Task

**Explanations of Various Error Types in LLMs Reasoning.** A manual review of the MountainCar task across three LLMs—GPT-3.5, Llama3-8b, and Llama3-70b—revealed significant differences in their explanations that were not necessarily anticipated from the quantitative analysis. Table 5 provides an overview of the types and Table 1 for counts of errors found in each model. During the evaluation, a single response could contain multiple error types. Despite Llama3-8b producing the shortest responses, it also had the highest error count.

(1) The first type of error, understanding the task, appeared frequently when the LLMs had to evaluate a proposed action, such as no acceleration in the MountainCar task. All three models tended to be concerned about overshooting the goal of reaching a position of $>= 0.5$. However, in this task, overshooting is irrelevant since the goal is to surpass 0.5. Similar replies across models suggest this mistake stems from a shared common-sense notion. Additionally, Llama3-8b often failed to recognise the presence of a hill on the left side.

(2) Logical mistakes were noted in GPT-3.5 and Llama3-70b when the LLMs justified moving left without recognising the need for oscillation to gain momentum, leading to paradoxical replies. These types of errors were more prevalent in Llama3-8b.

(3) Misunderstanding the history refers to the occasional misinterpretation or incorrect repetition of the history provided to the LLMs.

(4) Physical misunderstanding, though rare, involved incorrect responses regarding the effects of acceleration on velocity and similar cases.

(5) Mathematical errors commonly involved the LLMs disregarding the minus sign, leading them to believe that -0.5 is closer to 0.5 than 0.3. Although these mistakes led to awkward reasoning, they seldom significantly worsened the final decision.

(6) A common and human-like error involved judging when to switch directions to either gain or use momentum in the MountainCar task. Even the RL agent occasionally makes such mistakes.

Aside from the errors, GPT-3.5 demonstrated a better understanding of the task, often referring to the need to accelerate left to gain momentum for climbing the right hill. This was rarely mentioned by Llama3-70b and never by Llama3-8b, indicating GPT-3.5's superior task comprehension and explanatory ability. Llama3-70b, however, had an advantage in maintaining coherence, as it was less likely to contradict its arguments, unlike GPT-3.5, which occasionally argued against an action before ultimately supporting it. Both GPT-3.5 and Llama3-8b also displayed misunderstandings of the actions, such as incorrectly defining "action 0 (no acceleration)". This suggests a common-sense bias toward interpreting 0 as no action. Llama3-70b was better at retaining the task description in memory.

### G.1  A Compact Analysis of Error Types

Table 1 shows a quantitative analysis of the frequency of different error types committed by the LLMs for the MountainCar task. The evaluation highlighted various types of errors (see Table 5 in the Appendix), with Llama3-8b displaying the most errors despite its shorter responses. A common error among all models was misinterpreting the goal of the task, reflecting a shared common sense misunderstanding. Logical errors, particularly in oscillation movements, were prevalent in GPT-3.5 and Llama3-70b, while Llama3-8b frequently produced paradoxical replies. Misunderstanding the task history and physical principles was rare but present. Mathematical errors, especially disregarding the minus sign, occasionally impacted reasoning. Notably, GPT-3.5 demonstrated a better task understanding by referring to momentum strategies in the task, an insight less frequently or never mentioned by Llama3-70b and Llama3-8b, respectively. Llama3-70b did have one other advantage over other models as it was less often confused by its argument and excelled in maintaining task descriptions. Despite occasional errors in defining actions, GPT-3.5's superior comprehension of the task contributed to its higher-quality explanations.

## H  Guideline on Utilising LLMs for Agent Mental Modelling

LLMs are not yet capable of robust mental modelling. Below are some concrete suggestions for LLM and RL practitioners:

**Exercise caution.** In general, LLMs are prone to hallucination, and their reasoning should only be trusted after expert auditing to ensure a high rating of reliability/trustworthiness. It is crucial to explore ethical metrics to quantify trustworthiness.

**Consider multimodal input.** While our evaluation used only textual input, practitioners may consider incorporating multimodal environmental data (e.g., visual or sensory inputs), which could potentially enhance LLMs' understanding of behaviour and dynamics, leading to more trustworthy reasoning.

**Incorporate domain-specific RL data.** LLMs could benefit from action- and dynamics-related data (e.g., sequential/temporal trajectories) during pre-training or post-training stages to improve their ability to model RL agents. Practitioners may consider this practice when LLMs fall short of their expectation in mental modelling for tasks in their domain.

**Explore new learning paradigms.** Theoreticians may consider developing new learning paradigms that enables LLMs to learn and align with cause-effect relationships in input-output data, which could help LLMs perform agent mental modelling in a more trustworthy manner for a wider range of complex and unseen tasks.

