# OpenReview forum: "Mental Modelling of Reinforcement Learning Agents by Language Models"
_TMLR — Accepted by TMLR_

### Review · Reviewer_Eu2S · 2024-09-02

**Summary Of Contributions:**

This paper deals with the following questions: can modern LLMs, trained on vast amounts of world knowledge and possessing some already proven reasoning ability, be utilized to mentally model RL agents, and understand their actions as well as the environment they operate in? In this direction, the paper introduces the concept of agent mental modelling, which consists in building a mental model of the RL agent with the help of LLMs. The authors propose a specific evaluation framework that involves both actions as well as dynamics understanding, and detail a concrete template to interact with the LLM in-context and without re-optimizing or finetuning. The authors subsequently conduct an extensive experimental study on a variety of RL tasks of varying difficulty, where they investigate several aspects related to agent mental modelling. The evaluation studies in depth various practical parameters as well, such as history size, data format, data span and data variance, impact of indexing, error types etc. The authors reach the conclusion that LLMs are not yet fully capable of full agent mental modelling; in particular, despite their promise, pre-existing beliefs within LLMs (or general problems like hallucinations) prevent them from fully comprehending the agents' actions and behavior.

**Audience:**

Yes

**Broader Impact Concerns:**

No concern beyond what I mentioned previously.

**Claims And Evidence:**

Yes

**Requested Changes:**

I would be eager to change my score, but I first want to understand the following;
- Why did the authors not experiment with GPT-4o (except for the error type experiment)? It is possible that the more recent, state-of-the-art LLMs would have shown superior mental modelling capabilities. To me, the bigger problem is as follows: the authors state that "LLMs are not yet capable of fully realizing the mental modelling of agents through inference alone without further innovations."  Is it possible that this will not be as big of a problem with more recent LLMs such as GPT-4o? At least this is what the current Table 1 hints at?
- Is the evaluation framework + template introduced in this work valid for any MDP? It seems to me that the target of this work is primarily deterministic MDPs with deterministic policies. I cannot immediately see how the proposed evaluation framework could be used for arbitrary MDPs and arbitrary RL policies. Have the authors thought about experimenting with such MDPs?
- If LLMs are not yet capable of a robust mental modelling, then what is the takeaway message for practitioners? Use LLMs with a lot of caution? This work demonstrates that LLMs can very often be wrong, and can even produce a variety of distinct error types. Should practitioners then use them, and how exactly?

Even though it is not as critical, I would appreciate it if the authors could provide clarifications to the following:
- Can they elaborate more on the span, as I detailed above?
- I was not able to follow Appendix F.7. For step index 100, as the history size increases, the mismatch count also increases (from 0 to 7). But then how do the authors conclude that for larger steps, increasing history size minimally affects the LLMs’ predictions? I misinterpreted that figure, so additional clarifications could help.

**Strengths And Weaknesses:**

Strengths
- The paper studies an important problem for RL practitioners, which is motivated by the emergence of LLMs. Given LLMs can build powerful internal models of the world, it is logical to wonder whether they can help in the mental modelling of RL agents. This work makes a concrete step in this direction.
- The authors propose a detailed evaluation framework with a clear template. It does not only consider actions understanding but also dynamics understanding. Given the LLM is asked to provide information about its reasoning process, this allows to get a glimpse into the mental model of the LL  with respect to the RL Task at hand.
- The experimental study is detailed and involves several tasks (from easy to very hard ones). The task selection is interesting because an LLM could in principle use common-sense knowledge (built from vast knowledge bases) to reason about the tasks. The authors analyze various aspects (impact of history length, data format, data span and data variance,  impact of indexing, error types etc.). The empirical evaluation involved significant manual/human work, especially as far as the error type analysis is concerned.
- The classification of error types is interesting and sheds light on the internal reasoning inconsistencies of LLMs, and how these can affect RL agents.

Weaknesses
- I found it surprising and even confusing that the authors do not present extensive empirical results with GPT-4o, as they do with the other three LLMs. Table 1 already hints at the ability of GPT-4o to significantly reduce the total number of errors. It is well understood in the LLM literature that recent models (usually but not always with more parameters) can outperform older models. I was thus not able to understand why the authors performed such an extensive study using the three LLMs but omitting GPT-4o (with the exception of the experiment on error types, where GPT-4o was included). To me this is not simply a question of not including the most recent LLMs; the bigger question is whether the more recent LLMs would corroborate the current findings, or whether they would show much better performance. Is it for instance possible that the recent state=of-the-art LL:s could be much more successful in the mental modeling of RL agents, given they may have much better models of the world, common-sense facts/reasoning, etc.?
- The evaluation framework and, in particular, the action understanding and dynamics understanding presented in this work seem to make more sense for deterministic MDPs.
    - If a given action at a given state results in a distribution of next states (as opposed to a unique next state), then the LLM may fail to predict the next state, even if it outputs a valid state, i.e., a state appearing in the distribution over next states.
    - If there are multiple actions resulting in a given next state from a given start state, then the LLM may output any of them and still be correct.
- How to deal with the above point is not obvious at all. One idea is to restrict our interest to the space of deterministic MDPs with deterministic policies. If we want to cover general MDPs, we may want to output a distribution over states or actions, and possibly compute the KL-divergence between the ground truth and the LLM output? There are multiple possibilities here, and it is not clear to me what exactly the authors had in mind.
- I am not so sure that this work has a clear takeaway message to its readers. The work claims that their extensive empirical study shows that LLMs are not yet fully capable of realizing the mental modelling of agents, This is for sure an important finding. But then what is the author's suggestions? How exactly should LLMs be used? Perhaps they should not be used at all? Or be used with a great deal of caution? My concern is that there is not a very clear guideline or general takeaway message for LLM and RL practitioners and theoreticians. At the end of  the day, the work must male concrete suggestions based on its findings.
- I was not personally clear how Figures 19 was produced. My understanding is that each point in the plot corresponds to one state variable; state variables with a bigger space then get a higher span value of the x-axis. But how exactly are these values normalized to the 0-1 scale for each task. Is this done linearly? Some details there would have helped. Also, Furthermore, if the span is the real problem, would it have helped to simple rescale all state variables in each task, so that they all have the same high span? In that case, perhaps the variance would have been the main issue?

---

> ### Author Response · Authors · 2024-10-02
> **Rebuttal by Authors**
>
> We thank the reviewer for the detailed, constructive review and insightful comments.
>
> Responses to weakness (W) and requested changes (RC) are below (in multiple pages due to space limit):
>
> > **W1. Lacking the full test with gpt-4o model (SOTA) which would probably be more successful in the mental modeling of RL agents**
>
> We initially excluded GPT-4o from the full evaluation due to the high cost of running comprehensive tests. However, we did conduct preliminary experiments with GPT-4o on the MountainCar and Acrobot tasks, where it demonstrated an improved understanding of performance in actions and dynamics. In the revised version, we have extended the results to include GPT-4o's performance in predicting actions and dynamics across four control tasks (relevant figures have been updated accordingly).
>
> To highlight, GPT-4o exhibited significantly improved dynamics understanding compared to other models. Overall, the GPT-4o results further support our current finding that *LLMs, especially state-of-the-art models, are much more effective in mental modeling of RL agents compared to less performant models*. However, despite the overall improvement in dynamics understanding, the high-dim state spaces still pose a challenge for GPT-4o. For example, its dynamics prediction accuracy in the LunarLander task remains below 20%.
>
> We are finalising updates to all figures and tables with GPT-4o results, including those in the appendix.
>
> > **W2 & W3. The proposed evaluation framework seems to make more sense for deterministic MDPs**
>
> We appreciate the reviewer’s insightful comment and for bringing this into discussion. This study aims to evaluate whether current LLMs are capable of mental modelling of RL agents across a variety of RL tasks, encompassing different state-action spaces (small/large, discrete/continuous) and sparse/dense rewards in MDPs. While we primarily experimented with deterministic transitions, we acknowledge the limitation of not testing non-deterministic MDPs and think that such dynamics could introduce additional challenges for LLMs.
>
> These dynamics might deteriorate LLM performance in dynamics understanding due to the introduction of unpredictable randomness in state transitions, requiring LLMs to reason over distributions rather than scalar values. We also recognise that different ground truths might exist for LLM outputs in non-deterministic settings, as also indicated by the reviewer.
>
> Although introducing non-deterministic settings would increase complexity, the results from our current comprehensive study (e.g., dynamics prediction accuracy below 20% in LunarLander for the SOTA model GPT-4o) already indicate significant challenges for LLMs in mental modelling.
>
> We leave the experimentation of that setting for future work, but provide hints at how to adapt our evaluation framework to non-deterministic MDPs with minor adjustments to the prompt templates and the computing of evaluation metrics. We have integrated these ideas in the *Potential Direction* section of the main paper.
>
> Lastly, it is worth noting that in current RL literature (including works focused on RL performance and interpretability), many papers do not address all RL aspects, including non-deterministic environments. For example, several RL papers validated their methods in deterministic environments, to name a few, such as MuJoCo environments used in works like [Dynamics-Aware Unsupervised Discovery of Skills](https://arxiv.org/pdf/1907.01657), [Image Augmentation Is All You Need: Regularizing Deep Reinforcement Learning from Pixels](https://openreview.net/pdf?id=GY6-6sTvGaf), and most Atari games and other OpenAI Gym environments in XRL papers like [Towards Interpretable Deep Reinforcement Learning with Human-Friendly Prototypes](https://openreview.net/forum?id=hWwY_Jq0xsN) and [ProtoX: Explaining a Reinforcement Learning Agent via Prototyping](https://proceedings.neurips.cc/paper_files/paper/2022/file/ae5bf4f35236240c9460e761c60fa53d-Paper-Conference.pdf)).

---

> ### Author Response · Authors · 2024-10-02
> **Rebuttal by Authors**
>
> Consecutive responses:
>
> > **W4. Lacking a clear guideline or a general takeaway message based on the finding "LLMs are not yet capable of a robust mental modelling"**
>
> We thank the reviewer for the feedback.
>
> This study serves as a preliminary investigation into the capabilities of LLMs for the mental modelling of RL agents. On one hand, our results show that LLMs can model agent behaviour and dynamics to a certain extent in simpler tasks (e.g., MountainCar). On the other hand, their predictions and reasoning become untrustworthy on more complex tasks (e.g., high-dimensional Fetch tasks), revealing both the strengths and, unfortunately, limitations of current LLMs in agent mental modelling.
>
> At this stage, LLMs clearly lack multiple aspects required for true understanding (e.g., distinct error types), and relying on them for mental modelling without oversight could be harmful. As a result, **our key takeaway** is that rather than relying on a fully LLM-driven (automated) evaluation system for agent mental modelling, domain experts are expected to be involved upfront to audit LLMs' reasoning and ensure reasonable predictability and reliability before presenting them to non-expert practitioners. While LLMs are untrustworthy on certain tasks, we remain optimistic about their potential , given the observed trend of improved performance in more advanced models (e.g., GPT-4o exhibits remarkable dynamics understanding compared to others).
>
> > **W5. Not personally clear how Figures 19 in Appendix was produced.**
>
> *My understanding is that each point in the plot corresponds to one state variable; state variables with a bigger space then get a higher span value of the x-axis.*
>
> -> Yes, your understanding is correct.
>
> Figure 19 illustrates the influence of state span (one out of other factors) on prediction accuracy for individual state elements within LLMs' dynamics understanding. To clarify how this was produced:
>
> - For each task, the state span **was not scaled** during analysis because LLMs process the raw values of state variables (although internal LLM mechanisms may involve implicit preprocessing). Therefore, the linear analysis was conducted using the **raw** state span and the prediction accuracy of the state variables/elements.
> - However, when plotting the relationship between state span and prediction accuracy (as represented by the slopes of lines and $R^2$ values), we used a **dummy state span** (0-1) to allow for a clearer comparison of the relationships fitted by each language model across various tasks, as shown in Figure 7 of the main paper.
>
> > **RC1. It is possible that the more recent, state-of-the-art LLMs (like GPT-4o) would have shown superior mental modelling capabilities ... Is it possible that mental modelling of agents will not be as big of a problem with GPT-4o?**
>
> We have now included partial GPT-4o performance results for behaviour and dynamics understanding (across four tasks) and updated Figures 4, 5, and 6 accordingly. From the revised results, we observe that GPT-4o demonstrates superior mental modelling capabilities compared to other models. However, despite its improved performance, GPT-4o still falls short in dynamics understanding for tasks with higher state dimensions, such as LunarLander, where state prediction accuracy remains below 20%. Nevertheless, GPT-4o outperformed other models across all four control tasks in terms of dynamics understanding.
>
> Corresponding discussions have been added to the main text to reflect these (figure) updates.
>
> > **RC2. Is the evaluation framework + template introduced in this work valid for any MDP?**
>
> We discussed in W2 that our evaluation framework can be adapted for non-deterministic MDPs with minor adjustments to the prompt template and evaluation computing.
>
> For tasks with non-deterministic dynamic transitions, we propose the following potential solution hints:
>
> - for discrete state spaces (e.g., [FrozenLake](https://gymnasium.farama.org/environments/toy_text/frozen_lake/)): LLMs could be prompted to output a distribution over possible next state values. Multiple queries may be needed to approximate the distribution accurately, with evaluation using metrics like KL-divergence (as suggested by the reviewer). The ground truth probabilities for possible next states can be extracted from the simulator. Alternatively, LLMs could rank possible next state values, and this ranking could be compared to the ground truth ranking derived from the state probabilities.
>
> - for continuous state spaces: A practical approach may involve quantising state values into bins and prompting LLMs to output a distribution (or ranking) over sampled values from each bin. For multi-dimensional states, distributions/rankings can be generated for each dimension separately.

---

> ### Author Response · Authors · 2024-10-02
> **Rebuttal by Authors**
>
> Consecutive responses:
> > **[continue] RC2. Is the evaluation framework + template introduced in this work valid for any MDP?**
>
> The same approach can be applied to predicting a distribution over actions. The difference lies in the ground truth, which for PPO would be the mean and variance of the Gaussian policy, and for DQN, the Q-values for all possible (discrete) actions. In continuous action spaces, LLMs could be queried to output a mean and deviation for a Gaussian distribution, and this can be compared to the learned (Gaussian) distribution of the PPO agent.
>
> One concern we identified is that generating a distribution over states/actions may require LLMs to have access to a sufficient amount of history data to form an *accurate* distribution.
>
> We have integrated these potential solutions and their outcomes into the *Potential Direction* section of the main text.
>
> > **RC3. If LLMs are not yet capable of a robust mental modelling, then what is the takeaway message for practitioners?**
>
> We suggest rather than using a fullly LLMs-driven evaluation system to do mental modelling of RL agents, domain experts are expected to be involved in the first place to audit LLMs' reasonings and guarantee reasonable predicatability of LLMs before being presented to non-expert practitioners. We have added this takeaway in the *Potential Direction* section and provided concrete suggestions in a newly created section in Appendix H: *Guidelines on Utilising LLMs for Agent Mental Modeling*.
>
> > **RC4. Can they elaborate more on the span, as I detailed above?**
>
> We have added a clarification in Appendix F.3 to elaborate on the state span used during the linear analysis and the production of Figure 19.
>
> > **RC5. I was not able to follow Appendix F.7**
>
> We have added a clarification to the figure's caption (Figure 33 in Appendix F.7) to explain what each bar represents:
>
> *Each bar (corresponding to a different history size) at each step index indicates a mismatch between the LLMs' prediction (based on that history size) and the action taken by the RL agent at that step.*
>
> For example, at step index 100, as the history size increases, the LLMs' predictions still do not match the agent's actions. This leads to the conclusion that *increasing the history size minimally affects the LLMs’ predictions*.

---

### Review · Reviewer_fLL5 · 2024-09-13

**Summary Of Contributions:**

This paper investigates the capability of LLMs to reason about an agent’s behaviour and its effect on states via in-context learning. A large empirical study is conducted, including various evaluation metrics. The results show that LLMs are not yet capable of fully mentally modelling agents, and the results highlight some bias issues, impact of parameters such as history size, task instruction, and data format. There is also a comparison to human evaluations for comparison purposes.

**Audience:**

Yes

**Broader Impact Concerns:**

The algorithm used does not pose any specific ethical concern other than those already implied by LLMs and data analysis techniques.

**Claims And Evidence:**

Yes

**Requested Changes:**

The literature and technical descriptions should be extended where possible, as the paper does not give much context and technical understanding currently.

In the results 5.1, it is stated that the LLMs can accurately predict agent behaviours, “surpassing the random guess baseline”. This is a relatively low standard for accuracy. This statement can be adapted.

The confirmation bias in LLMs section seems somewhat speculative. Maybe the authors can try to reduce the gap between the hypothesis and the data (which is only performance data). In other words, there are a few hypotheses, but there are currently no tests for them, any of which may improve the paper.

**Strengths And Weaknesses:**

Positive:
- relatively extensive empirical study
- a potentially high impact and (to the best of my knowledge) original work helping to understand how LLMs can model RL agents


Negative:

- not too many technical insights following from this paper
- limited technical descriptions
- limited literature study, not so much in reference count, but more in terms of the explanation and discussion.
- lack of detail in some of the explanations, for instance, the bias issues.
- The large list of results makes it difficult to assess their correctness and as the analyses are rather short and do not stay close to the data, they take a relatively big leap.

---

> ### Author Response · Authors · 2024-10-02
> **Rebuttal by Authors**
>
> We thank the reviewer for the constructive review and insightful comments.
>
> Responses to weakness (W) and requested changes (RC) are below:
>
> > **W1 & W2. not too many technical insights following from this paper & limited technical descriptions**
>
> We thank the reviewer for the feedback, we addressed this concern by moving detailed descriptions of the prompt template design and post-processing of LLM outputs (previously in Appendices B and C) into the newly created Section 3.3 in the main text, providing more technical insights.
>
> > **W3. limited literature study, not so much in reference count, but more in terms of the explanation and discussion**
>
> We thank the reviewer for the feedback, we have expanded the literature review to include a discussion of the connection between agent mental modelling and explainable RL for added context.
>
> > **W4. lack of detail in some of the explanations, for instance, the bias issues**
>
> We thank the reviewer for the feedback, we have extended the explanations for bias issues.
>
> > **W5. the large list of results makes it difficult to assess their correctness and as the analyses are rather short and do not stay close to the data, they take a relatively big leap.**
>
> We thank the reviewer for the feedback. We modified the figures (e.g., Figure 4) to make the connection between data and analysis clear and provided some additional explanations of our approach. Should there be further concrete gaps between data and analysis, we would also be willing to make further changes.
>
> > **RC1. The literature and technical descriptions should be extended where possible, as the paper does not give much context and technical understanding currently.**
>
> We appreciate the feedback. In response, we have moved technical details from Appendix B and C into Section 3.3 for clarity and expanded the literature review, including a new paragraph discussing the connection between agent mental modelling and explainable RL for added context.
>
> > **RC2. In the results 5.1, it is stated that the LLMs can accurately predict agent behaviours, “surpassing the random guess baseline”. This is a relatively low standard for accuracy. This statement can be adapted.**
>
> We appreciate the feedback. Our main point is that LLMs can utilise agent history to build a mental model of RL agents, but they are not yet highly effective for our purpose. In the revised version, we have toned down our initial praise.
>
> > **RC3. The confirmation bias in LLMs section seems somewhat speculative. Maybe the authors can try to reduce the gap between the hypothesis and the data (which is only performance data). In other words, there are a few hypotheses, but there are currently no tests for them, any of which may improve the paper.**
>
> We appreciate the reviewer’s suggestion. We acknowledge that multiple hypotheses may explain the observation that *LLMs sometimes propose more optimal actions than the RL agent, as seen in the increased accuracy in human evaluations*. And we changed our text to reflect the possibility of other explanations. However, we do belive that our results show that the confirmation bias is the most likely explanation.
>
> Fully confirming this hypothesis would require advanced tools and mechanisms to thoroughly analyse the underlying reasoning processes of LLMs. We recognise that this is an interesting but challenging topic which, unfortunately, falls outside the current focus of this paper.

---

### Review · Reviewer_au9G · 2024-09-17

**Summary Of Contributions:**

This paper investigates the capacity of large language models (LLMs) to construct mental models of RL agents, which is formed as “agent mental modeling”. To take advantage of LLMs for this agent behavior modeling is a novel direction. The study looks at how well LLMs can predict an agent’s actions and their effects on the environment states within a markov decision process setup.

The authors prompt LLAMA3 8B, 70B instruct and GPT 3.5 with histories of actions and states, to predict agents’ actions and environment states as a proxy for understanding environment dynamics. This is done on a range of tasks including MountainCar, Acrobot, LunarLander, Pendulum, FetchPush, FetchSlide and FetchPickAndPlace.

The paper provides insights into LLMs’ capabilities and limitations in building a mental model of agents. They show that LLMs exhibit some promise in constructing mental models of agents. However, their performance is notably and inconsistently affected by factors such as history length, task complexity, pretraining bias and input data format.

**Audience:**

Yes

**Broader Impact Concerns:**

There are not any particular concerns on the ethical implications of this work.

**Claims And Evidence:**

Yes

**Requested Changes:**

In section 3, it would be helpful to change “last” to previous” as “last” could be interpreted as the final step or action, potentially confusing the reader.

In section 3, could you clarify “Judging the next action that is given”? Why $y=0$ when there is an agreement with action $a_{t+1}$?

The paper could benefit from further addressing how predictions of actions/states contribute to understanding of an agent's mental model and environment dynamics.

In Figure 4, it would be helpful to put task names on the right side of the rows.

For the LunarLander task, do you have any intuition as to why LLAMA3 8B outperforms the 70B model significantly?

The term "matching rate" should be clearly defined in the paper. It can be characterized as the prediction accuracy.

Figure 5 text could be bigger. There seems to be enough space.

Table 1 provides a nice analysis. To improve its readability, there is sufficient space to include both the error type names and numbers in the first column.

**Strengths And Weaknesses:**

**Strengths:**

This work addresses a relatively unexplored area of agent mental modeling using LLMs.
The study is conducted on a diverse set of RL tasks, ranging from classic control problems to more complex robotic manipulation scenarios.

**Weaknesses:**

Conceptually, would there be a need for agent mental modeling if the RL agent itself is based on an LLM that already generates human-readable text for its actions? Addressing this aspect seems to be overlooked in the paper.

Predicting the next state or action does not necessarily reflect a true understanding of the agent mental model or underlying dynamics of the environment. A model might achieve accurate predictions for the wrong reasons.
For the mental modeling to be meaningful, it is crucial not only to predict states/actions, but also to provide interpretable and usable reasoning. Without understanding the reasons behind predictions, it seems like we are back at square one.

One limitation of this work is primarily focusing on pre-existing tasks which limits the evaluation scope and could impact the results as LLMs could have gotten biased during pretraining with some knowledge about these tasks or optimal behavior. It would be interesting to see a similar analysis on new sets of tasks where LLMs could potentially exhibit less bias.

The introduction does not motivate why LLMs would be particularly suited for elucidating RL agent behavior beyond their ability to generate human-readable text.

Error type (1), "task understanding," is rather broad. How is task understanding evaluated in this context?

---

> ### Author Response · Authors · 2024-10-02
> **Rebuttal by Authors**
>
> We thank the reviewer for the detailed review and insightful comments.
>
> Responses to weakness (W) and requested changes (RC) are below (in multiple pages due to space limit):
>
> > **W1. Conceptually, would there be a need for agent mental modelling when the RL agent is based on an LLM that already generates human-readable text for its actions?**
>
> We thank the reviewer for raising this interesting point.
>
> Most existing RL agents are not based on LLMs. If we were to use an LLM-based RL agent, additional design would be required for trial-and-error learning, whereas our approach (evaluation framework) is more general (pluggable) and applies to existing RL agents trained and deployed without LLM involvement.
>
> If the LLM were part of the RL agent, it might self-justify its decisions rather than objectively explaining the agent's behaviour through the generated human-readable text. Therefore, an independent LLM is essential for performing the (true) mental modelling of RL agents, including LLM-based ones (an interesting direction for future exploration). Moreover, in this study, we not only explain the actions of RL agents but also evaluate how well LLMs understand these actions.
>
> > **W2. Predicting the next state or action does not necessarily reflect a true understanding of the agent mental model or underlying dynamics of the environment ...**
>
> We agree that mental modelling indicates a comprehensive understanding of the agent, which can be quite chanllenging to examine thoroughly. However, we belive that predictability is a requirement for understanding, which is why we evaluated it among other evaluation metrics. Understanding, both quantitatively and qualitatively, is something very hard to define, therefore we tried to evaluate multiple metrics to see how far LLMs can go.
>
> We started with quantitive measures, but also qualitatively conducted a manual review of LLMs' reasonings (in Sec. 5.3), where LLMs were tasked with judging the rationale behind actions. This revealed various types of errors and reasoning inconsistencies of LLMs, showing how these can impact the mental modelling of RL agents.
>
> > **W3. Evaluation focuses on pre-existing tasks which could have biased LLMs' reasonings.**
>
> We thank the reviewer for raising this point.
>
> It is worth noting that LLMs' biases are more likely to stem from task descriptions or knowledge of optimal behaviour in textual form (as the reviewer indicated), rather than from the numerical representation of trajectories and transitions, which were used in our study for agent mental modelling.
>
> We recognised the potential impact of task familiarity on LLMs' understanding performance. In Section 5.4, we conducted experiments to explore this, including an ablation study where we removed task descriptions from prompt templates to assess how prior knowledge, such as task goals or optimal behaviour, influence LLMs' agent mental modelling.
>
> While testing LLMs on entirely new tasks (those unknown to the LLMs) is an interesting future direction that could enhance our theoretical understanding, it seems unrealistic to expect LLMs to model agents without any knowledge of the task, which is why this was considered outside the scope of the current study. However, in Appendix F.8, we provided ablation experiments where we tested LLM performance with reduced task-specific knowledge. While these do not involve completely novel tasks, we believe they provide useful insights into this issue.
>
> > **W4. The introduction does not motivate why LLMs would be particularly suited for elucidating RL agent behavior beyond their ability to generate human-readable text.**
>
> We thank the reviewer for the feedback.
>
> LLMs seem to be particularly effective at mediating between human users and ML models, leveraging their broad general knowledge. We believe this makes them well-suited for explaining RL agents and building mental models of their behaviour. This in the end does come down to the power of LLMs and their ability to generate explanations that are not only human-readable but also human-understandable, setting them apart from other explainability techniques.
>
> Moreover, most existing explainability techniques for RL fail to consider the agent’s learning process or interactions with the environment when generating explanations for agents. In contrast, LLMs, with their strong in-context learning capabilities, can interpret an agent's interaction history and digest task-specific knowledge, making them uniquely suited for agent mental modelling.
>
> We have updated the introduction to highlight other arguments explaining why LLMs are uniquely suited for elucidating RL agent behaviour.

---

> ### Author Response · Authors · 2024-10-02
> **Rebuttal by Authors**
>
> Consecutive responses:
>
> > **W5. Error type (1), "task understanding," is rather broad. How is task understanding evaluated in this context?**
>
> We thank the reviewer for the feedback.
>
> Error type (1), "task understanding" refers to instances where the LLMs' response was concerned about something that does not influence the task. For example, as detailed in Appendix G, in the MountainCar task, the LLMs incorrectly expressed concern about overshooting the target goal. Since the task ends once the goal is reached, overshooting is irrelevant. In this case, the LLMs misinterpreted the goal of the task, likely drawing from the knowledge of other car-related scenarios. Such misunderstandings were categorised as error type (1).
>
> **Responses to all Requested Changes**
>
> Regarding RC1 & RC4 & RC6 & RC7 & RC8, we thank the reviewer for the detailed suggestions to improve the clarity of this paper, and we have addressed them in the updated version:
>
> - RC1, we highlight the equivalence of "last" and "previous" when we first define "last" in section 3.2.
>
> - RC4, done with task names added to the figure's right side
>
> - RC6, highlighted when first introduced in section 3
>
> - RC7, Figure 5 updated with bigger text
>
> - RC8, done with the requested change
>
>
> > **RC2: In section 3, could you clarify “Judging the next action that is given”?**
>
> This evaluation question is designed to manually assess the reasoning of LLMs (as detailed in Section 5.3) when queried to evaluate the rationale behind agents' actions. Rather than directly asking LLMs why a specific action was taken by RL agents, we prompt them to *judge* the rationale behind the decision. This approach prevents the LLMs from automatically justifying the action and instead encourages a more critical and objective assessment.
>
> > **[continue]RC2: Why y = 0 when there is an agreement with action a_t+1?**
>
> We have to codify the LLMs' responses (agree/disagree) to facilitate automatic evaluation.
>
> > **RC3: The paper could benefit from further addressing how predictions of actions/states contribute to understanding of an agent's mental model and environment dynamics.**
>
> We believe that accurate predictions are essential for understanding. Evaluating the predictive capabilities of the model helps to determine how well it understands the agent's mental model. We have updated the method section 3 to include this motivation.
>
> > **RC5: For the LunarLander task, do you have any intuition as to why LLAMA3 8B outperforms the 70B model significantly?**
>
> As shown in Figure 4, the 70B model significantly outperforms the 8B model in terms of next and last action prediction match rates. However, for the next state prediction, where all models perform poorly, the 8B model shows a slightly higher matching rate (1-2% more than the 70B). We believe this is likely due to noise, as none of the models seem capable of grasping the dynamics of next/last state prediction in the LunarLander task. Therefore, in line with intuition, the 70B model generally outperforms the 8B model in LunarLander.

---

### Author Response · Authors · 2024-10-02
**Rebuttal by Authors**

## To all the reviewers: thanks for the reviews and a summary of key changes
We thank all the reviewers ($\textcolor{blue}{\text{Reviewer-Eu2S}}$, $\textcolor{red}{\text{Reviewer-fLL5}}$, $\textcolor{orange}{\text{Reviewer-au9G}}$) for their time and effort in reviewing our paper. The reviewers highlighted the following strengths:

**Highlighted strength**:

1. studied an important and well-motivated problem and made a concrete step in the direction of whether LLMs can help in the mental modelling of RL agents (by $\textcolor{blue}{\text{Reviewer-Eu2S}}$)
2. relatively extensive empirical study (by $\textcolor{red}{\text{Reviewer-fLL5}}$); conducted on a diverse set of RL tasks (by $\textcolor{orange}{\text{Reviewer-au9G}}$)
3. "LLMs are not yet fully capable of realizing the mental modelling of agents" is an important finding (by $\textcolor{blue}{\text{Reviewer-Eu2S}}$)
4. relatively unexplored area of (RL) agent mental modeling using LLMs (by $\textcolor{orange}{\text{Reviewer-au9G}}$)
5. an original work helping understand how LLMs can model RL agents, with a potentially high impact (by $\textcolor{red}{\text{Reviewer-fLL5}}$)

We have further improved the paper based on the valuable feedback from all reviewers, with changes highlighted in different coloured text corresponding to each reviewer’s comments.

**A summary of the key changes we made**:

1. updated figures to include GPT-4o (SOTA model) results ($\textcolor{blue}{\text{Reviewer-Eu2S}}$)
2. added more context to the introduction and related work sections ($\textcolor{orange}{\text{Reviewer-au9G}}$, $\textcolor{red}{\text{Reviewer-fLL5}}$)
3. added clarification to the methodology section ($\textcolor{orange}{\text{Reviewer-au9G}}$, $\textcolor{red}{\text{Reviewer-fLL5}}$)
4. expanded explanations/discussion in result section ($\textcolor{blue}{\text{Reviewer-Eu2S}}$, $\textcolor{red}{\text{Reviewer-fLL5}}$, $\textcolor{orange}{\text{Reviewer-au9G}}$)
5. added takeaway message and limitations ($\textcolor{blue}{\text{Reviewer-Eu2S}}$)

In our responses, "RC" refers to "Requested Changes" (e.g., RC1 for the first requested change), while "W" refers to weaknesses, with W1 indicating the first identified weakness.

We greatly appreciate the reviewers' efforts and valuable suggestions. We've carefully addressed each comment and made improvements accordingly. We hope these revisions resolve the concerns and welcome any further feedback.

---

### Decision · Action_Editor_CmKj · 2024-11-19

**Recommendation:** Accept with minor revision

**Comment:**

The paper studies an important problem for RL practitioners.
The authors propose a detailed evaluation framework with a clear template. It does not only consider actions understanding but also dynamics understanding. Given the LLM is asked to provide information about its reasoning process, this allows to get a glimpse into the mental model of the LLM with respect to the RL Task at hand.
The experimental study is detailed and involves several tasks (from easy to very hard ones). The task selection is interesting because an LLM could in principle use common-sense knowledge (built from vast knowledge bases) to reason about the tasks. The authors analyze various aspects (impact of history length, data format, data span and data variance, impact of indexing, error types etc.). The empirical evaluation involved significant manual/human work, especially as far as the error type analysis is concerned.

The paper will benefit from a section discussing the potential for future-proofing this approach, especially as more agents become LLM-based and can independently provide human-readable rationales before taking actions. This would add value to the discussion.

**Audience:**

Intersection of RL and LLM community, explainable RL

**Claims And Evidence:**

This paper empirically examines how well large language models  can build a mental model of reinforcement learning (RL) agents, termed agent mental modelling, by reasoning about an agent's behaviour and its effect on states from agent interaction history. This research attempts to unveil the potential of leveraging LLMs for elucidating RL agent behaviour. To this end, they propose specific evaluation metrics and test them on selected RL task datasets of varying complexity, reporting findings on agent mental model establishment. The results disclose that LLMs are not yet capable of fully realising the mental modelling of agents through inference alone without further innovations.